# UniDoc-Bench: A Unified Benchmark for Document-Centric Multimodal RAG

## ABSTRACT

Multimodal retrieval-augmented Generation (MM-RAG) is a key approach for applying large language models (LLMs) and agents to real-world knowledge bases, yet current evaluations are fragmented—focusing on either text or images in isolation, or simplified multimodal setup, failing to capture document-centric multimodal use cases. In this paper, we introduce UniDoc-Bench, the first large-scale, realistic benchmark for MM-RAG built from 70k real-world PDF pages across 8 domains. Our pipeline extracts and links evidence from text, tables, and figures, then generates $1,600$ multimodal QA pairs spanning factual retrieval, comparison, summarization, and logical reasoning queries. To ensure reliability, 20% of QA pairs are validated by multiple annotators and expert adjudication. UniDoc-Bench supports apples-to-apples comparison across four paradigms — 1) text-only, 2) image-only, 3) *multimodal* text–image fusion and 4) *multimodal* joint retrieval — under a unified protocol with standardized candidate pools, prompts, and evaluation metrics. Our experiments show that multimodal text–image fusion RAG systems consistently outperform both unimodal and jointly multimodal embedding–based retrieval, indicating that neither text nor images alone are sufficient and that current multimodal embeddings remain inadequate. Beyond benchmarking, our analysis reveals when and how visual context complements textual evidence, uncovers systematic failure modes, and offers actionable guidance for developing more robust MM-RAG pipelines.

## 1 INTRODUCTION

Retrieval-augmented generation (RAG) has become a widely used approach for applying large language models (LLMs) and agents to real-world knowledge bases (Gao et al., 2023; Fan et al., 2024). The dominant text-only pipeline applies Optical Character Recognition (OCR) (Li et al., 2022; Xue et al., 2024; Poznanski et al., 2025) to flatten document pages into text, indexes them as chunks, retrieves top-k text passages, and feeds them to a generator. However, many answers depend on information embedded in figures, charts, tables, and complex layouts, where OCR often discards crucial spatial and visual semantics (e.g., map, axes, bar lengths, color encodings) (Ma et al., 2024a; Faysse et al., 2024a). These limitations have driven the rapid development of multimodal RAG (MM-RAG), which embeds documents across modalities (text, tables, and images) and retrieves and reasons over them jointly, emerging as a key paradigm for document intelligence.

Current MM-RAG evaluation benchmarks exhibit substantial limitations, as summarized in Table 1. Many are restricted to a single image or a single document page as reference (Mathew et al., 2021; 2022; Zhu et al., 2022; Li et al., 2024; Ma et al., 2024b), cover narrow domains Mathew et al. (2021; 2022); Zhu et al. (2022); Li et al. (2024), under-represent modalities (Li et al., 2024; Mathew et al., 2022), operate at limited scale (few queries/pages) (Ma et al., 2024b; Wang et al., 2025b) or lack a highly relevant database for RAG evaluation (Ma et al., 2024b). These gaps hinder fair and comprehensive comparison across methods. Moreover, debatable claims have emerged — such as that "image retrieval is all you need" (Faysse et al., 2024a; Su et al., 2025) or that multimodal retrieval is inherently superior (Zhang et al., 2024b; Yu et al., 2024b)— without enough fair and unified evaluation. In response, we introduce UniDoc-Bench, a manually verified benchmark spanning 8 domains and covering text, chart, and table content, explicitly designed for cross-modality grounding with examples shown in Figure 1. Crucially, UniDoc-Bench enables apples-to-apples evaluation of text-retrieval, image-retrieval, multimodal text-image-fusion retrieval, and multimodal joint

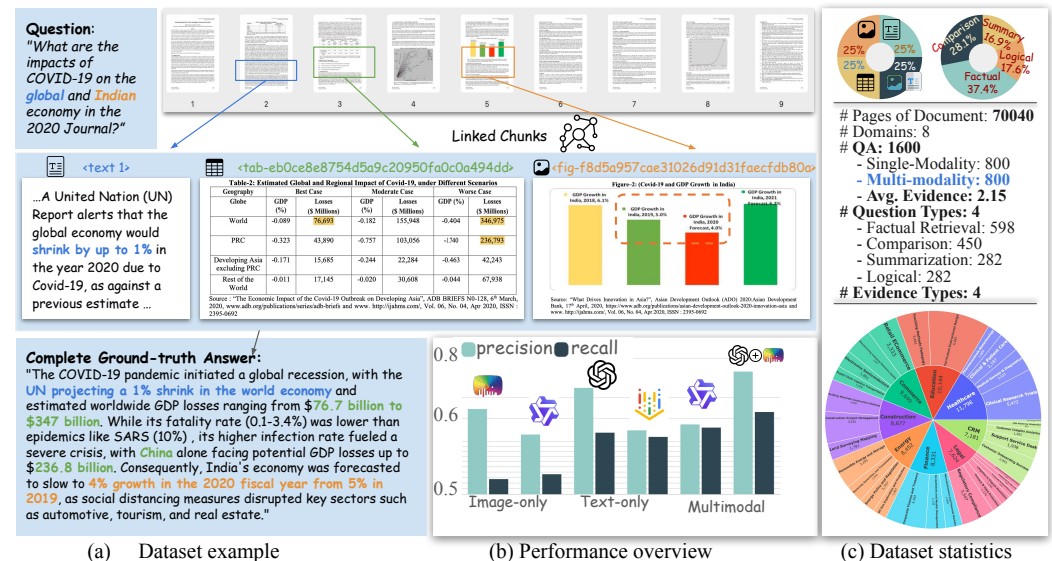

Figure 1: `UniDoc-Bench` overview.

retrieval pipelines using highly relevant large document database and multi-type, cross-modality-grounding queries under a unified protocol. This setup provides an unbiased view of when multimodal retrieval offers advantages beyond single modalities. In practice, `UniDoc-Bench` quantifies multimodal gains, guides system design choices, and accelerates the development of effective MM-RAG systems for real-world document intelligence.

We curate a high-quality multimodal RAG evaluation benchmark by designing and applying a classification-based filtering scheme to unlabeled, real-world PDF documents (PDFA (Montalvo & Wightman, 2024)), yielding 70k highly relevant pages across eight widely used domains — *Finance, Legal, Healthcare, Commerce and Manufacturing, CRM, Energy, Education, and Construction*—containing rich cross-modality content, including text, tables, and images. We construct a knowledge graph that links cross-modality contents across documents via overlapping entities, and leverage these connections to synthesize 1,600 QA pairs spanning four question types: *factual retrieval*, *comparison*, *summarization*, and *logical reasoning*, enabling multi-modality grounding and reflecting realistic retrieval scenarios. To ensure quality, 20% of the QA pairs are evaluated by three independent annotators for faithfulness, completeness, self-containment, human intent, and evidence usability, with disagreements resolved through expert adjudication. Figure 2 illustrates the full pipeline from PDF segmentation to dataset creation and evaluation.

In this paper, we compare text-only, image-only, multimodal joint, and text-image-fusion retrieval augmented generation pipelines under a unified setup, using identical candidate pools, fixed top-$k$, consistent prompts, and standardized evaluation criteria. We report retrieval metrics (`Recall@10`, `Precision@10`), answer `completeness` and `faithfulness` defined at Section 4.2. We observe consistent gains for text–image-fusion RAG systems (`completeness` = 68.4%) over multimodal joint retrieval systems (64.1%), text-retrieval systems (65.3%), and image-retrieval systems (54.5%). This indicates that retrieving text and images separately using dedicated embeddings, then combining them in the final LLM query, outperforms unified embeddings or single-modality retrieval. Moreover, visual evidence improves answer completeness and enhances faithfulness when paired with textual context, though image-only retrieval cannot fully capture the textual information contained in images. Questions requiring images to answer remain challenging for all systems, suggesting that future RAG improvements should prioritize image-dependent queries. In contrast, performance differences across question types, such as comparison or factual retrieval, are minimal.

In this paper, we make the following contributions:

- We introduce a new multimodal RAG benchmark built from real-world PDF documents, comprising 70k pages across 8 domains, with 1,600 human-verified QA pairs referencing text, figures, and tables, spanning 4 question types.

Table 1: Comparison of existing dataset with `UniDoc-Bench`.

| Benchmarks | Domain | Evidence | # Queries | # Pages of Doc | RAG Suitable | Unified Evaluation | Multiple Reference |
|---|---|---|---|---|---|---|---|
| ArxivQA (Li et al., 2024) | single | 🖼 | 0.5k | - | ✗ | ✗ | ✗ |
| TAT-DQA (Zhu et al., 2022) | single | 📄 ▦ | 1.6k | - | ✗ | ✗ | ✗ |
| InfoVQA (Mathew et al., 2022) | single | 🖼 | 0.5k | - | ✗ | ✗ | ✗ |
| DocVQA (Mathew et al., 2021) | single | 🖼 ▦ | 0.5k | - | ✗ | ✗ | ✗ |
| MMLONG (Ma et al., 2024b) | multiple | 📄 🖼 ▦ | 1k | 6k | ✗ | ✗ | ✓ |
| REALMM (Wasserman et al., 2025) | multiple | 📄 🖼 ▦ | 5k | 8k | ✓ | ✗ | ✗ |
| ViDoSeek (Wang et al., 2025b) | multiple | 📄 🖼 ▦ | 1.2k | 10k | ✓ | ✗ | ✗ |
| `UniDoc-Bench` (ours) | multiple | 📄 🖼 ▦ | 1.6k | 70k | ✓ | ✓ | ✓ |

**RAG Suitable**: The dataset provides RAG-style data: queries are self-contained and reflect realistic human questions, with each paired to a grounding corpus (text, images, tables) for retrieval-conditioned answering, supported by a large, highly relevant knowledge base to evaluate retrieval. **Unified Evaluation**: Apples-to-apples comparison across different baseline RAG systems. **Multiple Reference**: Supports multi-hop, multi-modality, multi-source grounding.

- We present an associated data synthesizing pipeline for creating multimodal RAG evaluation datasets, designed to be compatible with any document database.
- We propose a fair and reproducible evaluation framework by fixing candidate pools across modalities , and measuring retrieval effectiveness, answer faithfulness, and completeness end-to-end across different RAG systems. Specifically, to ensure fairness when comparing against text-only RAG, we caption images and tables and match them back to the retrieved text chunks before final generation, thereby maintaining a consistent candidate pool.
- We conduct a systematic comparison of text-retrieval, image-retrieval, text–image fusion, and multimodal joint retrieval pipelines, analyzing which retrieval strategy performs best under different question types, evidence modalities, and document characteristics, providing practical guidance for choosing MM-RAG systems in real-world data settings.

## 2 RELATED WORKS

### 2.1 MULTIMODAL RETRIEVAL-AUGMENTED GENERATION (MM-RAG)

Recent advances in multimodal understanding highlight the importance of MM-RAG in reducing hallucinations. VLM2Vec (Jiang et al., 2024; Meng et al., 2025) demonstrated that instruction-tuning vision-language models significantly enhances their ability to produce robust embeddings, leading to strong performance across diverse text–image alignment tasks. Similarly, SeBe (Chen et al., 2025) adapts LLaVA-1.5 (Liu et al., 2024) by finetuning it into a retrieval-oriented embedding model, aligning user queries with external knowledge sources. GME (Zhang et al., 2024a) proposed a unified multimodal embedding model that is able to perform both text-to-image, image-to-text, and text-to-text retrieval. Uni-Retrieval (Jia et al., 2025) extends the paradigm by integrating VLMs with prompt-tuning strategies, enabling flexible handling of heterogeneous queries and modalities. Routing-based methods such as UniversalRAG (Yeo et al., 2025) and UniRAG (Sharifymoghaddam et al., 2025) introduce adaptive query routing mechanisms that dynamically select the most appropriate modality and level of granularity.

### 2.2 VISUAL DOCUMENT UNDERSTANDING AND EVALUATION

The challenge of document understanding with interleaved textual and visual components has recently prompted the development of specialized vision-based RAG pipelines (Yu et al., 2024a; Wang et al., 2025a;c) that directly take screenshots of documents as input. A notable example is ColPali (Faysse et al., 2024a), which leverages VLMs to jointly encode textual queries and visual documents with the MaxSim operations (Khattab & Zaharia, 2020). ViDoRAG (Wang et al., 2025a) introduces a multi-agent reasoning architecture designed for complex queries that require iterative cross-modal reasoning. In parallel, optimization-focused approaches such as VRAG (Wang et al., 2025c) apply reinforcement learning strategies, including GRPO-based Shao et al. (2024) training, to adapt VLMs for end-to-end document understanding. However, the comparisons with text-only baselines are not entirely fair, as most of these baselines exclude non-text modalities in response generation. Moreover, existing evaluations are conducted on datasets not designed for RAG. MMLongBench-Doc (Ma et al., 2024c) targets long-context multimodal document understanding,

Figure 2: Data Construction pipeline. (a) We filter and tag PDFA documents to curate a high-quality database of 70k pages spanning 8 domains. (b) We parse documents into text, figures, and tables, then synthesize initial QA pairs covering four question types and three modalities using adapted templates. (c) We ground answers in supporting evidence, refine questions for human-intent and self-containment, and verify responses for factuality and completeness, yielding $1,600$ QA pairs. To ensure quality, $20\%$ of the dataset is validated by three independent human annotators.

but its database is not highly relevant and thus unsuitable for retrieval tasks. REAL-MM (Wasserman et al., 2025) and VidoSeek (Wang et al., 2025b) are designed for MM-RAG, yet they lack cross-modality and multi-page evidence, limiting their ability to provide comprehensive and unified evaluation across RAG systems. The other benchmarks (Mathew et al., 2021; 2022; Zhu et al., 2022; Li et al., 2024) are typically limited to a single image or a single document page, covering narrow domains, under-representing modalities, or operating at limited scale with only a few queries or pages, as summarized in Table 1. To address these gaps, we introduce `UniDoc-Bench`, a benchmark tailored to practical RAG use cases.

## 3 DATASET CURATION

First, a large-scale, high-quality multi-modal database is needed for evaluating RAG systems, where each document contains content-rich figures, tables and corresponding textual information. Documents should be domain-specific and exhibit high inter-document similarity to evaluate effective retrieval. The construction of this database is detailed in Section 3.1. Then, we require high-quality query–answer pairs to evaluate the RAG system. Each query is designed to reflect realistic human intent and is written as a self-contained question. The corresponding ground-truth answer must be retrievable solely from the curated database and supported by evidence across multiple modalities. In Section 3.2, we describe our synthetic QA pipeline, and in Section 3.3, we validate dataset quality through human annotation.

### 3.1 SOURCE DOCUMENT COLLECTION

We use PDFA (Montalvo & Wightman, 2024) as our data source, containing diverse formats (e.g., reports, slides, posters) and covering broad domains, but it lacks tags or labels. Therefore, our first step is data filtering to collect a high-quality database. We design a field scheme (see Appendix B.1) that captures key metadata, including domain, subdomain, language, modality (e.g., text, tables, figures), image quality (whether the resolution is clear), and text proportion. This allows us to standardize the data and build a high-quality cross-modality database. As shown in Figure 1 (c), we select 8 domains based on differences across industries and define many subdomains within each, grouping similar documents. To ensure high inter-document similarity, we retain only documents from 3–5 related subdomains containing multiple modalities, yielding on average $8,000$ pages per domain. The final dataset spans *Legal, Commerce and Manufacturing, Education, Energy, Construction, Finance, Healthcare, and CRM*, with detailed subdomain descriptions in Appendix B.2.

### 3.2 QUESTION AND ANSWER SYNTHESIS PIPELINE

As shown in Figure 2, we introduce a data-synthesis pipeline for building multimodal RAG evaluation datasets with high-quality QA pairs, compatible with various document databases.

#### 3.2.1 EVIDENCE COLLECTION

**PDF Parsing.** We first parse our curated PDF document database[1] by extracting text chunks, tables, and figures, with the latter two stored separately as image files. Within the parsed text

---

[1] https://unstructured.io/

chunk, each image and table is replaced with a unique placeholder tag (e.g., `<<fig-XXX>>` or `<<tab-XXX>>`), along with its corresponding caption and parsed content to fully represent interleaved multimodal content. An example of this parsing process is provided in Appendix B.3.

**Chunks Grouping.** To support multimodal evidence QA, we construct a knowledge graph ($\mathcal{G}_i$) (ExplodingGradients, 2024; Peng et al., 2024) over the parsed chunks for domain $i$, where nodes ($N_i = \{n_{i1}, n_{i2}, ...\}$) represent chunks and edges ($E_i$) denote overlapping entities (e.g., "AI Agent Platform"). Chunks across three modalities (text, tables, figures), from within or across documents, are linked to form ground-truth evidence, which are then used for QA synthesis in the next step.

### 3.2.2 Question and Answer Generation

**Template Choice.** First, we ensure the synthesized questions are **diverse** and span multiple categories, since focusing on a single category or using only the same few-shot example questions can introduce bias and limit the comprehensiveness of RAG evaluation. We designed four RAG question types: 1) `factual retrieval`, 2) `comparison`, 3) `summarization`, and 4) `logical reasoning`. For each question type and document domain, we design 10–15 general templates (see Appendix B.4). We then sample linked chunks ($n_{ij}, e_{ij}, n_{ik}$) and prompt the LLM to select 1–3 templates ($T_{ij}$) that best match the provided chunks and are most likely to produce QA pairs that humans would naturally ask, thereby improving both the diversity and coverage of the questions.

**Evidence Grounding.** To ensure comprehensive evaluation of MM-RAG, we design four *answer types* with distinct evidence requirements, each supported by specialized prompts:

- Text-only: The question can be fully answered using natural language text from the documents.
- Image-only: The question requires information exclusively from an image, such as numerical values shown only in a figure, thereby testing the system's ability to interpret visual content.
- Image-plus-text: Answering the question requires integrating information from both text and images, testing the model's ability to reason across modalities.
- Table-required: The question required tabular information to answer, requiring the system to understand table structure and content.

To construct QA pairs, we prompt `GPT-4.1` with parsed text chunks and extracted figures/tables (PNG format), guided by prompts $P_n$ corresponding to the above answer types (see details in Appendix B.5) and templates $T_{ij}$. We then employ `Gemini-Pro-2.5` — to mitigate single-LLM bias — to verify that the ground-truth answers are correctly grounded in the referenced text, tables, or images, ensuring factual correctness and re-classifying question types when necessary.

**Rewriting.** To ensure that questions are **self-contained** and reflect realistic **human intent**, we refine the initially synthesized QA pairs. In the first stage, many synthesized questions follow a long-context QA style and may include vague references such as "in this report" or "in Figure 8." To make them suitable for RAG evaluation, we rewrite these questions to ensure they are self-contained and understandable without external context (Appendix B.6). Additionally, many QA pairs are grounded in images, leading to VQA-style questions (e.g., "How many logos are in Apple Inc.'s 2023 report?"). Such questions do not reflect natural human queries in a RAG context, so we filter and rewrite them to better align with realistic human intent. To ensure comprehensive evaluation, ground-truth answers must be **complete** and **diverse**. In the final step, we revise answers to cover all relevant aspects of their corresponding questions (see Appendix B.7).

**Deduplication and Balance.** Additionally, we remove duplicated question–answer pairs that are highly similar in the question or the answer (similarity $> 0.75^2$) to maintain dataset quality and diversity. We also rebalance the dataset by question type and answer type to provide a fair evaluation.

**Dataset Statistics.** Based on the above stages, we construct an evaluation benchmark consisting of 200 QA pairs for each category, in total 1600 QAs as described in Section 3.1. Within each set of 200 QA pairs, we maintain an equal distribution of 50 text-only, image-only, text-plus-image, and table-only questions. In total, the dataset contains 800 single-modality and 800 multi-modality questions. On average, each question requires 2.15 evidence items (text chunks, images, or tables) for a complete answer, highlighting the need for RAG systems to retrieve multiple pieces of evidence. We further ensure a balanced distribution across the four main question types: factual retrieval, summarization, comparison, and logical reasoning. More details can be found in Figure 1(b).

---

[2]https://huggingface.co/sentence-transformers/all-MiniLM-L6-v2

Table 2: Human evaluation quality on a 20% sample ($n$=320). Each cell shows % and (count/320)

| | Factuality–Q | Factuality–R | Completeness | Self-Contained | Human-like Intent | Grounding |
|---|---|---|---|---|---|---|
| **% & Count** | 99.70% (319/320) | 91.90% (294/320) | 91.90% (294/320) | 99.70% (319/320) | 97.50% (312/320) | 84.38% (270/320) |

## 3.3 DATASET QUALITY

We evaluate whether our constructed dataset is of sufficient quality to support reliable evaluation of different RAG systems by sampling 20% of our dataset— 40 QA pairs from each domain, resulting in a total of 320 QA pairs—for human evaluation. We recruited 3 human annotators to evaluate the question–response pairs against the provided source documents. In cases where 3 annotators disagreed, a 4th senior reviewer mediated the discussion and guided the annotators toward a consensus decision. For each item, annotators were directed to a folder containing all relevant source materials, including text extracted from PDF documents and associated images. The annotation process involved assessing each question-response pair across five dimensions (More details about this human annotation task can be found in Appendix C):

- **Factuality**: evaluates whether the claims made in the question (`Factuality-Question`) and the response (`Factuality-Response`) were factually supported by the source documents.
- **Completeness**: assesses whether the response incorporates all necessary information from the retrieved sources to fully answer the question.
- **Grounding**: assesses whether each source chunk (text, image, or table) used to generate the ground-truth response is necessary to answer the question, by labeling it as either `required` or `not required`.
- **Self-Contained**: assesses whether the question was understandable and answerable on its own, without needing external context beyond the provided documents.
- **Human-like Intent**: evaluates whether the question reflected a natural, meaningful query that a human would plausibly ask to retrieve information.

As shown in Table 2, the sample shows near-perfect question factuality and self-containment, with strong response factuality and completeness (each ≈294/320). Human-like intent remains high (312/320). Grounding label accuracy is solid (270/320) as well.

## 4 EXPERIMENTS

To fairly evaluate different RAG systems, we focus on two aspects: retrieval and end-to-end performance. In this section, we first evaluate the retrieval performance of four embedding and retrieval models, including text-only, image-only, and two multimodal approaches (Section 4.1). We then assess the end-to-end response performance of six RAG systems that vary in their use of embeddings, retrieval strategies, and LLMs (Section 4.2). Together, these experiments highlight the utility of our dataset and provide practical guidance for selecting RAG components.

### 4.1 RETRIEVAL PERFORMANCE

**Baselines.** We use the curated PDF documents as the knowledge base and the synthesized QA pairs to evaluate 4 embedding–retrieval models. For all methods, we retrieve the top-$k = 10$ candidates.

- **Text:** PDF pages are parsed into text chunks, each embedded with OpenAI's `text-embedding-3-small`, and retrieved via vector search.
- **Image:** Each PDF page is converted to a JPEG image, which is embedded using `ColQwen2.5-v0.2` (Faysse et al., 2024b) for image retrieval.
- **MM:** Both text chunks and page-level images are embedded.
  - MM (`GME`): Text and images are jointly embedded using `GME-Qwen2-VL-7B-Instruct` (Zhang et al., 2024a), enabling multimodal retrieval.
  - MM (T+I): A fusion baseline that selects the top-5 candidates from Text and the top-5 from Image retrieval.

**Metrics.** We report `Precision@10` and `Recall@10` as the retrieval metrics. Since no re-ranker is applied, recall is more informative than `nDCG` for evaluation. Since we need to evaluate both image and text retrieval, each retrieved text chunk or PDF image-page is mapped back to its original

Table 3: Retrieval performance (`Precision@10` / `Recall@10`) of 4 RAG systems on 1600 QA pairs across eight domains, with average recall reported across all domains.

| Domain | Text (OpenAI) | | Image (`colqwen`) | | Multimodal | | | |
| | | | | | GME | | Text + Image | |
| | *Precision* | *Recall* | *Precision* | *Recall* | *Precision* | *Recall* | *Precision* | *Recall* |
|---|---|---|---|---|---|---|---|---|
| Com. | 0.430 | 0.813 | 0.294 | 0.831 | 0.354 | **0.895** | **0.523** | 0.886 |
| Constr. | 0.377 | 0.750 | 0.263 | 0.794 | 0.336 | **0.881** | **0.451** | 0.833 |
| CRM | 0.400 | 0.808 | 0.283 | 0.829 | 0.343 | **0.884** | **0.486** | 0.876 |
| Edu | 0.414 | 0.843 | 0.268 | 0.843 | 0.366 | **0.912** | **0.460** | 0.880 |
| Energy | 0.382 | 0.772 | 0.257 | 0.822 | 0.257 | 0.822 | **0.459** | **0.863** |
| Fin. | 0.384 | 0.778 | 0.291 | 0.812 | 0.376 | 0.857 | **0.484** | **0.867** |
| HC | 0.420 | 0.741 | 0.252 | **0.849** | 0.370 | 0.835 | **0.460** | 0.837 |
| Legal | 0.440 | 0.864 | 0.291 | 0.855 | 0.327 | 0.876 | **0.510** | **0.891** |
| Avg. | 0.406 | 0.796 | 0.275 | 0.829 | 0.341 | **0.870** | **0.479** | 0.867 |

PDF page, and the ground-truth contexts are mapped in the same way. Consequently, a retrieved chunk may span multiple consecutive pages of the source document (e.g., pages 2–3 of document A). A retrieval is considered a true positive if the retrieved text chunk or image-page matches the ground-truth context in both page number and file. This criterion may slightly inflate `Recall@10`, since partial overlaps (e.g., retrieved pages 1–3 vs. ground-truth pages 3–5, with the answer on page 5) are still treated as correct. However, this approach offers the most practical and fair basis for comparing text and image retrieval. Thus, absolute scores should not be overinterpreted; the key is the relative performance differences across methods, which remain reliable.

Table 3 reports the retrieval performance of the four RAG embedding-retrieval models. We observe that **image-based retrieval achieves consistently higher recall but lower precision than text-based retrieval**, as page-image chunks cover more information than individual text chunks. Combining text and image retrieval (T+I) further improves both recall and precision, effectively leveraging the strengths of both modalities. In contrast, multimodal embeddings (`gme-Qwen2-VL-7B-Instruct`), which encode text and images jointly rather than separately, achieve comparable recall but substantially lower precision, suggesting that current multimodal embeddings still lag behind fusion of unimodal embeddings. We also break down retrieval performance by question and answer types in Appendix E.1.

## 4.2 END-TO-END PERFORMANCE

**Baselines.**

- **Image-only RAG:** Each PDF page is converted to a JPEG and retrieved via image embeddings.
    - **Image-only RAG** (IMG): Uses LlamaIndex with `colqwen2.5-v0.2` (Faysse et al., 2024b) for image retrieval and `GPT-4.1` as the final MM-LLM. Each PDF page is converted to a JPEG image and embedded. After retrieval, the question and retrieved images are provided to `GPT-4.1` to obtain the final response.
    - **VRAG** (Wang et al., 2025d): a multimodal RAG agent that leverages a vision-specific action space—including operations such as cropping and scaling—to iteratively extract information from image-formatted PDF pages in a coarse-to-fine manner. The embedding model is `colqwen2.5-v0.2`, and the final LLM is `GPT-4.1`.
- **Text-only RAG:** Most multimodal RAG studies (Wang et al., 2025b; Faysse et al., 2024a) compare only against text-only baselines. For a fairer comparison, PDF pages are parsed into text chunks, embedded for retrieval, with associated images/tables linked back for final responses.
    - **TEXT**: Each text chunk is embedded using `text-embedding-3-small` and retrieved. The retrieved text chunks, along with their associated images, are then fed into `GPT-4.1` to generate the final response.
    - **Vertex AI**: following the official tutorial[3], PDFs are parsed into text and images, with images auto-captioned by Gemini. Only the text (document text and image captions) is indexed by `text-embedding-004` and retrieved, and the retrieved chunks along with the corresponding images are passed to `gemini-2.5-flash` for final response.

---

[3]https://www.cloudskillsboost.google/focuses/85643?parent=catalog

Table 4: Completeness of six RAG systems on 1,600 QA pairs across eight domains. Average recall is reported across all domains, with similarity top-$k$ set to 10 and 20, computed against the ground-truth responses.

| Domain | Image-only RAG | | | | Text-only RAG (+img matched) | | | | Multimodal RAG | | | |
| | IMG | | VRAG | | TEXT | | Vertex AI | | MM (GME) | | T+I | |
| | top-10 | top-20 | top-10 | top-20 | top-10 | top-20 | top-10 | top-20 | top-10 | top-20 | top-10 | top-20 |
|---|---|---|---|---|---|---|---|---|---|---|---|---|
| Com. | 0.545 | 0.552 | 0.547 | 0.550 | 0.633 | 0.673 | 0.613 | 0.630 | 0.617 | 0.611 | **0.693** | **0.733** |
| Constr. | 0.502 | 0.601 | 0.536 | 0.542 | 0.561 | 0.587 | 0.558 | 0.621 | **0.616** | 0.609 | 0.607 | **0.647** |
| CRM | 0.524 | 0.524 | 0.523 | 0.544 | 0.643 | 0.663 | 0.628 | 0.625 | 0.623 | 0.637 | **0.647** | **0.703** |
| Edu | 0.569 | 0.560 | 0.517 | 0.524 | **0.692** | **0.702** | 0.613 | 0.633 | 0.640 | 0.668 | 0.688 | 0.691 |
| Energy | 0.535 | 0.566 | 0.558 | 0.589 | 0.607 | 0.637 | 0.627 | 0.677 | **0.669** | 0.666 | 0.649 | **0.680** |
| Fin. | 0.500 | 0.499 | 0.529 | 0.535 | 0.584 | 0.626 | 0.557 | 0.605 | 0.627 | 0.636 | **0.638** | **0.636** |
| HC | 0.481 | 0.492 | 0.481 | 0.492 | 0.602 | 0.639 | **0.638** | 0.643 | 0.642 | 0.664 | 0.621 | **0.666** |
| Legal | 0.558 | 0.568 | 0.599 | 0.595 | 0.629 | 0.696 | 0.642 | 0.675 | 0.609 | 0.629 | **0.689** | **0.716** |
| **Avg.** | 0.527 | 0.545 | 0.536 | 0.546 | 0.619 | 0.653 | 0.610 | 0.639 | 0.630 | 0.641 | **0.654** | **0.684** |

- **MM-RAG:** Both text chunks and image-format page images are embedded and retrieved for responses.
    - **Multimodal Text-Image-Fusion RAG (T+I):** Retrieves text and images separately using `text-embedding-3-small` and `colqwen2.5-v0.2`, then combines them for generation with `GPT-4.1`.
    - **Multimodal-joint-Retrieval RAG** (MM): Uses `gme-Qwen2-VL-7B-Instruct` (Zhang et al., 2024a) as a multimodal embedding model for both text and image content. Unlike T+I, where text and images are embedded and retrieved separately, the text chunks and image-formatted PDF pages are embedded together, retrieved jointly, and then fed into `GPT-4.1` for the final response.

**Metrics.** For **end-to-end** performance, we use an LLM-based judge to measure faithfulness and completeness. Specifically, we first ask the LLM to extract the facts required to answer each question and then verify whether these facts are grounded in the ground-truth chunks; this is measured as `faithfulness`. Next, we ask the LLM to extract the facts required to answer the question from the ground-truth answer and then check whether each fact appears in the system's response; this is measured as `completeness`. Higher faithfulness and completeness scores are better.

Table 4 reports the completeness of responses generated by the six RAG systems under varying similarity top-$k$ retrieval settings. **Text-only RAG** (0.653) **substantially outperforms Image-only RAG systems** (IMG: 0.545, VRAG: 0.546), highlighting the significant performance gap between text-based and image-based retrieval in current RAG architectures. Although image retrieval achieves higher completeness at the retrieval stage, this advantage does not translate into better end-to-end performance, since multimodal LLMs (GPT-4.1) are more effective when processing text and image chunks together rather than page-level image PDFs alone. The text-image-fusion RAG achieves the best overall performance (0.684) across eight domains, demonstrating that image-based PDF representations can effectively complement text retrieval. Although VRAG leverages cropping and scaling to enhance image-based retrieval (0.536 for VRAG vs. 0.527 for IMG (top$k = 10$)), it still lags behind the combined Text&Image-Retrieval approach, underscoring the advantage of explicitly integrating both modalities. Multimodal joint-retrieval RAG systems (MM; 0.641) also fall short of the simple combination of the best text and image embeddings. This indicates that current multimodal embedding approaches still have substantial room for improvement, and that explicitly **combining separate text and image embeddings remains the most effective strategy** for leveraging multimodal documents. More notably, multimodal-joint RAG (MM; 0.641) performs worse than text-only RAG (0.653), demonstrating that current multimodal models still fall short of strong unimodal baselines. These results also highlight the importance of establishing fair baselines and the value of our dataset: multimodal RAG systems should be benchmarked against strong, balanced baselines on diverse and high-quality datasets rather than against overly weak text-only settings.

Table 5 shows that questions requiring only text are most effectively handled by RAG systems with text-embedding. **Questions requiring tables are also relatively easy for RAG systems**, as tables can be accurately parsed as text, which is a straightforward step before embedding documents for text-based retrieval. In contrast, questions requiring images remain challenging across all embed-

Table 5: Faithfulness and Completeness of six RAG systems across different question and answer types on 1,600 QA pairs spanning eight domains, with average recall reported across all domains.

| Type | Image-only RAG | | | | Text-only RAG (+img matched) | | | | Multimodal RAG | | | |
|---|---|---|---|---|---|---|---|---|---|---|---|---|
| | IMG | | VRAG | | TEXT | | Vertex AI | | MM (GME) | | T+I | |
| | *faith.* | *complet.* | *faith.* | *complet.* | *faith.* | *complet.* | *faith.* | *complet.* | *faith.* | *complet.* | *faith.* | *complet.* |
| F.R. | 0.640 | 0.536 | 0.581 | 0.536 | 0.698 | 0.629 | 0.563 | 0.557 | 0.668 | 0.599 | **0.763** | **0.704** |
| Comp. | 0.669 | 0.510 | 0.611 | 0.513 | 0.739 | 0.619 | 0.634 | 0.644 | 0.744 | **0.656** | **0.755** | 0.641 |
| Summary | 0.727 | 0.536 | 0.706 | 0.602 | 0.736 | 0.613 | 0.694 | **0.670** | 0.752 | **0.670** | **0.781** | 0.651 |
| Logical | 0.738 | 0.526 | 0.650 | 0.584 | 0.769 | 0.607 | 0.690 | 0.660 | 0.744 | **0.678** | **0.780** | 0.621 |
| Text-only | 0.812 | 0.580 | 0.767 | 0.624 | 0.877 | 0.656 | 0.817 | 0.758 | 0.849 | **0.771** | **0.880** | 0.700 |
| Img-only | 0.512 | 0.448 | 0.453 | 0.483 | 0.580 | 0.606 | 0.359 | 0.447 | 0.463 | 0.436 | **0.620** | **0.615** |
| Text + Img | 0.678 | 0.498 | 0.576 | 0.523 | 0.716 | 0.601 | 0.581 | 0.556 | 0.707 | 0.583 | **0.749** | **0.630** |
| Table-req. | 0.693 | 0.587 | 0.662 | 0.554 | 0.714 | 0.601 | 0.716 | 0.670 | **0.819** | **0.747** | 0.811 | 0.716 |

ding types — text, image, or multimodal — highlighting that future **RAG improvements should prioritize image-required questions**. We also observe that **question type has minimal impact on overall RAG performance**. We provide detailed case studies in Appendix D.

### 4.3 ADDITIONAL FINDINGS

**MM-RAG systems can offer both improved end-to-end performance and lower cost compared to text-only RAG.** As reported in Appendix E.3, text-only RAG is the most expensive, image-only RAG has the lowest cost and latency, and multimodal RAG is cheaper than text-only RAG while maintaining comparable latency.

**Open-source and commercial multimodal embeddings perform comparably.** We compare RAG systems using different multimodal embeddings (Table 8, Table 9) and find that the commercial `voyage-multimodal-3` achieves similar performance to the open-source `GME`, though both still lag behind multimodal text–image fusion RAG systems.

**Content-rich images increase difficulty.** We classify images using `gemini-2.5-pro` as content-rich (containing information not in the text) or illustrative. Content-rich images are more prevalent in finance (62.8%) and construction (69.3%) than in commerce manufacturing (40.0%) and legal (49.5%), indicating that domains with more content-rich images pose greater challenges for RAG, consistent with the results in Table 4. Details are in Appendix F.1 .

**Question type affects difficulty.** We further analyzed fined-grained evidence types and found that RAG performance depends on answer modality: text retrieval excels at entity recognition (53.9% better than image retrieval), comparative analysis (37.6%), contextual numerical reasoning (34.8%), and quantity estimation (29.1%), while image retrieval is stronger on chart/table interpretation (64.2% better than text retrieval), temporal trends (40.0%), and spatial/geographic reasoning (13.3%). Detailed examples and analysis are in Appendix D.1, Appendix D.2 and Appendix F.2.

We also summarize in Appendix F that single document page numbers and formats do not significantly affect MM-RAG performance.

## 5 CONCLUSION

In this paper, we introduced `UniDoc-Bench`, a large-scale benchmark for document-centric multimodal RAG, built from 70k real-world PDF pages across 8 domains with $1,600$ human-verified QA pairs. Our experiments establish a clear performance hierarchy, showing that **text-image fusion RAG performs the best**, consistently outperforming both joint multimodal (MM) RAG and single-modality RAG systems. This key finding demonstrates that fusing separate, strong retrievers for text and images is currently a more effective strategy than relying on a single joint multimodal embedding or a single modality alone. Our analysis further pinpoints image-dependent queries as the primary challenge for all systems. By providing a standardized platform for fair comparison, `UniDoc-Bench` serves as a crucial resource to guide the development of more robust and faithful document intelligence systems.

ETHICS STATEMENT

In our paper, we strictly follow the ICLR ethical research standards and laws. To the best of our knowledge, our work abides by the General Ethical Principles.

REPRODUCIBILITY STATEMENT

We adhere to ICLR reproducibility standards and ensure the reproducibility of our work. All source datasets we employed are publicly available (PDFA). We are making our code available in the supplementary materials to enable replication of our findings.

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

## A  THE USE OF LARGE LANGUAGE MODELS (LLMS)

We used LLMs for three purposes: (i) polishing grammar and improving readability, and (ii) assisting in the evaluation of RAG outputs (iii) synthesizing the QA pairs. All research ideas and analyses were conducted by the authors, who take full responsibility for the content.

## B  DATASET CREATION DETAILS

### B.1  DOCUMENT FIELDS

We classify each PDF document into the following fields:

- *domain*: one or more from {Healthcare, Finance, Technology and Software, Commerce and Manufacturing, Marketing, Arts and Entertainment, Government, Legal, Education, Scientific Research and Development, Customer Relationship Management (CRM). others}
- *subdomain*: optional finer-grained categories
- *date*: year or estimated year (e.g., 2005)
- *language*: language of the document (e.g., en)
- *modality*: possible values include {text, table, figure, formula, image, drawing}
- *quality*: parsing confidence, values {easy-parse, hard-parse}
- *format*: one or more from {form, report, notice, paper, slide, poster, book, newspaper, article, textbook, note, webpage, document, record}
- *text_proportion*: percentage of textual content (e.g., 25%)

As described in Section 3.1, we do not include every domain or subdomain in our benchmark. Instead, we filter the source data and retain eight highly representative domains.

### B.2  DOMAIN DEFINITIONS

We classify documents into domains and subdomains, each with a brief description for clarity. These labels are used for tagging. As detailed in Section 3.1, we filter the source data and retain eight highly representative domains rather than including all possible ones.

| Domain | Subdomain | Description |
|---|---|---|
| Healthcare | Clinical & Patient Care | Direct provider-patient interaction: diagnosis, treatment, and care management. |
| Healthcare | Pharmaceuticals & Biotechnology | Development and regulation of drugs, vaccines, and biotechnological products (no patient records). |
| Healthcare | Medical Devices & Diagnostics | Design, production, and regulation of medical equipment and diagnostic tools (no patient records). |
| Healthcare | Clinical Research & Trials | Controlled studies testing treatments, drugs, or therapies. |
| Healthcare | Public Health & Policy | Population-level promotion, disease prevention, accessibility (not individual records). |
| Healthcare | Other Healthcare Topics | Healthcare economics, law, and alternative medicine. |
| Finance | Investments & Wealth Management | Stock portfolios, retirement planning, mutual funds, hedge funds. |
| Finance | Insurance & Risk Management | Health, life, auto, property insurance; actuarial analysis. |
| Finance | Corporate Finance & Treasury | Budgeting, fundraising, M&A, investor relations, corporate structure. |

| Domain | Subdomain | Description |
|---|---|---|
| Finance | Personal Finance & FinTech | Budgeting apps, personal loans, P2P lending, digital wallets. |
| Finance | Real Estate Finance | Mortgages, REITs, valuations, market dynamics. |
| Finance | Macroeconomics & Financial Markets | Markets, currency, fiscal/monetary policy, global economics. |
| Finance | Other Finance Topics | Microfinance, Islamic banking, niche financial products. |
| Technology & Software | Software Engineering & DevOps | Coding, testing, deployment, CI/CD, APIs. |
| Technology & Software | Cybersecurity & Information Security | Risk management, encryption, compliance, network defense. |
| Technology & Software | Data Science, AI & Analytics | ML, pipelines, visualization, BI tools. |
| Technology & Software | HCI & UX | Design, prototyping, accessibility, usability studies. |
| Technology & Software | Emerging Technologies | AR/VR, quantum computing, IoT, blockchain. |
| Technology & Software | Other Tech Topics | Legacy systems, databases, systems architecture. |
| Commerce & Manufacturing | Supply Chain & Logistics | Procurement, warehousing, transportation, inventory. |
| Commerce & Manufacturing | Industrial Engineering & Production | Process optimization, quality control, Lean/Six Sigma. |
| Commerce & Manufacturing | Retail & E-Commerce | Marketplaces, POS systems, consumer engagement. |
| Commerce & Manufacturing | Trade Policy & Global Commerce | Tariffs, export-import regulation, global trade. |
| Commerce & Manufacturing | Other Commerce Topics | Business operations, sales, distribution. |
| Marketing | Digital Marketing & Advertising | Social media, SEO/SEM, online campaigns. |
| Marketing | Consumer Behavior & Market Research | Surveys, focus groups, data-driven insights. |
| Marketing | Branding & Corporate Identity | Logo, image, brand value, messaging. |
| Marketing | Marketing Analytics & Metrics | ROI, attribution models, dashboards. |
| Marketing | Other Marketing Topics | Public relations, sponsorships, offline campaigns. |
| Arts & Entertainment | Performing Arts | Music, theater, dance, performance reviews. |
| Arts & Entertainment | Visual Arts & Design | Painting, sculpture, illustration, graphic design. |
| Arts & Entertainment | Film, TV & Media Studies | Criticism, production, audience reception. |

| Domain | Subdomain | Description |
|---|---|---|
| Arts & Entertainment | Literature & Writing | Fiction, non-fiction, literary analysis. |
| Arts & Entertainment | Games & Interactive Media | Video games, role-playing, esports. |
| Arts & Entertainment | Other Arts Topics | Fashion, photography, cultural heritage. |
| Government | Public Administration & Policy | Bureaucracy, policymaking, implementation. |
| Government | Law Enforcement & Security | Policing, intelligence, defense, military studies. |
| Government | International Relations & Diplomacy | Foreign policy, treaties, global governance. |
| Government | Elections & Governance | Voting, political systems, representation. |
| Government | Other Government Topics | Civil rights, immigration, taxation. |
| Legal | Corporate & Business Law | Contracts, mergers, compliance. |
| Legal | Criminal & Civil Law | Courts, trials, disputes, legal rights. |
| Legal | Intellectual Property Law | Copyrights, patents, trademarks. |
| Legal | International & Comparative Law | Cross-border legal systems, treaties. |
| Legal | Legal Theory & Jurisprudence | Philosophy of law, frameworks. |
| Legal | Other Legal Topics | Niche legal issues, regulatory law. |
| Education | K-12 Education | Curriculum, pedagogy, assessments. |
| Education | Higher Education & Academia | Universities, research, accreditation. |
| Education | Online & Distance Learning | MOOCs, e-learning, virtual platforms. |
| Education | Education Policy & Reform | Accessibility, standards, funding. |
| Education | Other Education Topics | Lifelong learning, teacher training. |
| Scientific R&D | Natural Sciences | Physics, chemistry, biology, earth science. |
| Scientific R&D | Engineering & Applied Sciences | Electrical, mechanical, civil, aerospace. |
| Scientific R&D | Medical & Life Sciences | Biomedical, genetics, ecology. |
| Scientific R&D | Computer Science & Computational Fields | Algorithms, theory, AI, networks. |
| Scientific R&D | Other Science Topics | Interdisciplinary, niche fields. |
| CRM | Customer Support & Helpdesk | Call centers, chatbots, support tickets. |
| CRM | Sales & Lead Management | CRM tools, customer tracking, pipelines. |
| CRM | Customer Analytics & Insights | Segmentation, lifetime value, churn analysis. |

| Domain | Subdomain | Description |
|---|---|---|
| CRM | Customer Experience (CX) & Engagement | Feedback, personalization, loyalty programs. |
| CRM | Other CRM Topics | Partnerships, integrations, omni-channel strategies. |

### B.3 PARSING EXAMPLES

We use `unstructured` to parse each PDF into three components: text chunks, images of figures, and images of tables. Since many figures (e.g., signatures or logos) are not informative, we only retain figures that include captions. Figure 3 shows an example of the parsing output, where figures are represented by placeholders such as `<<fig-XXX>>` and the parsed text from the figures.

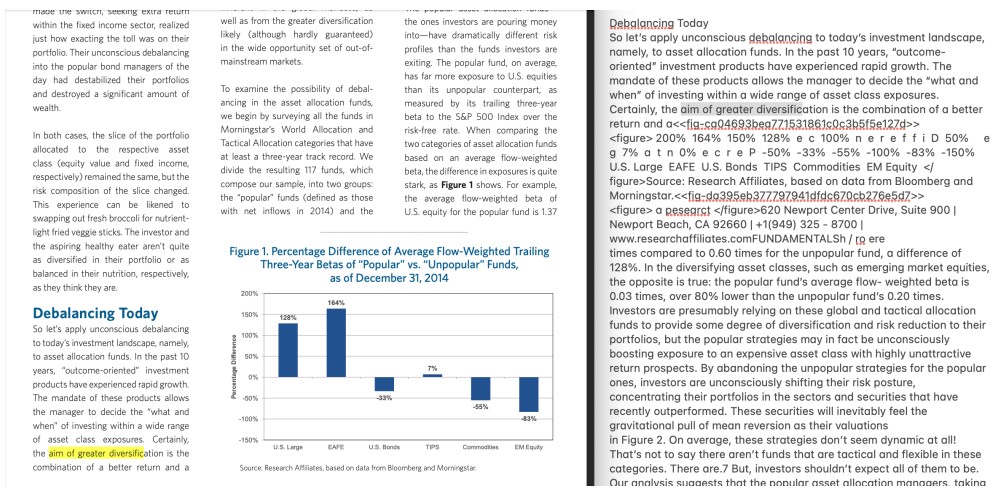

Figure 3: Example of PDF parsing with figure placeholders (`<<fig-XXX>>`).

## B.4 DATASET TEMPLATES

This is the templates for the domain: `finance`. We create different templates for different domains, which can be found in our code files in the supplementary materials.

### FACTUAL RETRIEVAL

| Template | Example |
|---|---|
| What indicators, policies, or tools are described in the discussion of [Economic Topic/Financial Strategy]? | What inflation indicators are cited in the ECB's policy blog from June? |
| Which markets, sectors, or instruments are emphasized in relation to [Trend/Event/Goal]? | Which sectors are favored in the 2025 sustainable investing outlook? |
| What key positions or exposures are taken by [Investor/Desk/Division] in response to [Condition/Event]? | What position changes did the multi-asset team make in response to rising real yields? |
| What assumptions, constraints, or parameters are specified in [Scenario/Strategy/Model]? | What assumptions are used in the stress testing scenario for oil price shocks? |
| When was [Policy/Event/Adjustment] implemented, and what immediate actions followed? | When did the Bank of Japan change its yield curve control stance? |
| Who oversees or initiates [Financial Decision/Policy/Investment Move] in the described context? | Who approves short-term borrowing requests in the global treasury function? |
| How is [Strategy/Instrument/Term] defined or operationalized in this context? | How is "duration-neutral tilt" defined in the Q3 fixed income note? |
| How do you carry out or execute [Action/Transaction/Plan] in [Financial Context]? | How do you implement a covered call overlay in an income-focused portfolio? |
| What are the procedural steps or controls listed for [Financial Task/Compliance/Change]? | What steps are required to evaluate bond ladder rollovers in rising rates? |

### COMPARISON

| Template | Example |
|---|---|
| How do [Strategies/Regions/Instruments] compare in terms of [Risk/Performance/Conditions]? | How do TIPS and gold compare for inflation protection in the current macro setup? |
| Which asset class, sector, or product is better suited for [Objective/Environment]? | Which is better for income stability in retirement: dividend ETFs or bond ladders? |
| What are the structural or tactical differences between [Financial Approaches]? | What are the key differences between liability-driven investment and balanced allocation strategies? |
| How did [Metric/Position/Exposure] change between [Period A] and [Period B]? | How did corporate cash allocation to floating-rate debt shift over 2023? |
| How do regulatory or monetary responses differ between [Jurisdictions]? | How does Fed liquidity provision compare to ECB emergency facilities post-crisis? |

### SUMMARIZATION

| Template | Example |
|---|---|
| What are the key findings or takeaways from [Brief/Update/Policy/Strategy]? | What are the key points in the tactical asset allocation update from July? |
| Summarize the main market movements, themes, or risks discussed in [Note/Newsletter/Memo]. | Summarize the interest rate risk themes highlighted in the October bond outlook. |
| What portfolio, liquidity, or policy adjustments are recommended or implemented? | What rebalancing steps were taken in the client model portfolios in Q1? |
| List the major economic risks or opportunities discussed in [Period/Event/Note]. | What macro risks are cited ahead of the U.S. election cycle? |
| What are the key operational or structural features of [Product/Plan/Tool]? | What are the structural features of the new drawdown facility described in the treasury toolkit? |

CAUSAL / REASONING / WHY QUESTIONS

| Template | Example |
|---|---|
| Why did [Entity/Desk/Advisor] make [Move/Shift/Decision] in response to [Condition/Event]? | Why did the balanced portfolio reduce international equity in Q2? |
| How did [Macro Event/Regulatory Shift] influence [Positioning/Allocation/Operations]? | How did the Basel III revisions alter corporate liquidity buffers? |
| What drove the shift from [Approach A] to [Approach B] in [Context]? | What drove the shift from risk-parity to volatility-targeting in multi-asset allocation? |
| Why was [Instrument/Policy/Vehicle] introduced or phased out? | Why was the internal netting structure retired in the 2024 treasury overhaul? |
| What sequence of factors or events led to [Market Reaction/Portfolio Impact/Policy Result]? | What sequence of events led to capital outflows from EM debt in late 2023? |

## B.5 QA SYNTHESIZING PROMPTS

### B.5.1 TEXT-ONLY

---

**Prompt P.1: Text-only RAG Question Generation**

Prompt: You are an assistant specialized in creating Multimodal RAG tasks. The task is the following: Given some natural language contexts and images inside these contexts, you will generate questions that can be asked by a user to retrieve information from a large documentary corpus.

**Requirements:**

- The 2-hop synthesized question must be a single, self-contained question and must not use "and" to connect multiple questions.

- The answer of the synthesized question will only be found in the contexts.

- The answer of the synthesized question cannot be found in the images.

- The synthesized question must require all the chunks in the contexts to be answered.

- The synthesized question must be specific enough to locate the contexts in a large documentary corpus.

- You must also provide an explanation why the answer can only be found in the provided contexts.

**Question Template:**

- Use the following template to generate the QA:

  ```
  {{TEMPLATES}}
  ```

**Output Format:**

```
{
    "questions": [
        {
            "question": "<synthesized-question>",
            "answer": "<answer-of-the-question>",
            "question_type":
            <choose from "factual_retrieval", "comparison",
                "summarization", "causal_reasoning">,
            "explanation-chunks": "<explanation-chunks>",
            "sentences-chunks-used": {"Chunk1": "sentences-chunk1",
                "Chunk2": "sentences-chunk2", ...}
        }
    ]
}
```

**Input Data:**

- Contexts: "{{contexts}}"

- Images: The image is as follows:

**Notes:**

- If the image can only be used for visualization or illustration, return an empty list for 'sentences-chunks-used'.

- If you cannot use all the chunks in the answer, return an empty list for 'sentences-chunks-used'.

---

### B.5.2 IMAGE-ONLY

---

**Prompt P.2: Image-only RAG Question Generation**

Prompt: You are an assistant specialized in creating Multimodal RAG tasks. The task is the following: Given some natural language contexts and images inside these contexts, you will generate questions that can be asked by a user to retrieve information from a large documentary corpus.

**Requirements:**

1. The synthesized question must be a single, self-contained question and must not use "and" to connect multiple questions.

2. The answer of the synthesized question will only be found in the image and cannot be found in any sentences in the chunks of the provided contexts.

3. The synthesized question must require chunks/contexts to locate the image and cannot mention the image directly.

4. The synthesized question must be specific enough to locate the contexts in a large documentary corpus.

5. Do not ask "what XYZ in the graph/image/figure"; the question must be general enough to be asked in a large corpus.

6. If you cannot synthesize a question which can only be answered in the image based on the above requirements, do not synthesize anything.

7. Provide an explanation why the answer can only be found in the image and cannot be found in the provided chunks/contexts.

8. Avoid phrasing like "what is shown in the image," e.g., "what color/logo/name in the image."

9. Emphasize reasoning, aggregation, temporal comparison, or retrieval from source data. Imagine the question being asked without the image still making partial sense.

**Question Template:**

- Use the following template to generate the QA:

```
{{TEMPLATES}}
```

**Output Format:**

```
{
    "questions": [
        {
            "question": "<synthesized-question>",
            "answer": "<answer-of-the-question>",
            "question_type":
            <choose from "factual_retrieval", "comparison",
                "summarization", "causal_reasoning">,
            "image": "<<fig-aaaaa>>",
            "explanation-image": "<explanation-image>",
            "explanation-chunks": "<explanation-chunks>",
            "sentences-chunks-used":
            {"Chunk1": "sentences-chunk1",
                "Chunk2": "sentences-chunk2", ...}
        }
    ]
}
```

**Input Data:**

- Contexts: "{{contexts}}"

- Images: The image is as follows:

**Notes:**

- If the image can only be used for visualization or illustration, return an empty list for 'sentences-chunks-used'.

- If you cannot use all the chunks in the answer, return an empty list for 'sentences-chunks-used'.

---

### B.5.3 TEXT-PLUS-IMAGE

---

**Prompt P.3: Text-plus-image RAG Question Generation**

Prompt: You are an assistant specialized in creating Multimodal RAG tasks. The task is the following: Given some natural language contexts and images inside these contexts, you will generate questions that can be asked by a user to retrieve information from a large documentary corpus.

**Requirements:**

1. The 2-hop synthesized question must require both the provided contexts and images to answer.

2. The concise answer of the synthesized question will directly require information in the image to answer.

3. The concise answer of the synthesized question will also require information in the natural language contexts to answer.

4. The synthesized question must require contexts to locate the image and cannot mention the image directly.

5. The synthesized question must be specific enough to locate the contexts in a large documentary corpus.

6. Provide an explanation indicating which part of the image is used to answer and which sentence in the contexts is used to answer the question.

7. Do not ask "what XYZ in the graph"; the question must be general enough to be asked in a large corpus.

8. If you cannot synthesize a question based on these requirements or directly use the information in the images, do not synthesize anything.

9. If the image can only be used for visualization or illustration, do not synthesize anything. If you cannot use all the chunks in the answer, do not synthesize the question.

10. The synthesized question must be a single, self-contained question and must not use "and" to connect multiple questions.

**Question Template:**

• Use the following template to generate the QA:

```
{{TEMPLATES}}
```

**Output Format:**

```
{
    "questions": [
        {
            "question": "<synthesized-question>",
            "answer": "<answer-of-the-question>",
            "question_type": <choose from "factual_retrieval",
                "comparison", "summarization", "causal_reasoning">,
            "image": "<<fig-aaaaa>>",
            "explanation-image": "<explanation-image>",
            "explanation-chunks": "<explanation-chunks>",
            "sentences-chunks-used":
            {"Chunk1": "sentences-chunk1",
                "Chunk2": "sentences-chunk2", ...}
        },...
    ]
}
```

**Input Data:**

• Contexts: "{{contexts}}"

• Images: The image is as follows:

**Notes:**

• If the image can only be used for visualization or illustration, return an empty list for 'sentences-chunks-used'.

• If you cannot use all the chunks in the answer, return an empty list for 'sentences-chunks-used'.

### B.5.4 TABLE-REQUIRED

---

**Prompt P.4: Table-required RAG Question Generation**

Prompt: You are an assistant specialized in creating Multimodal RAG tasks. The task is the following: Given some natural language contexts containing tables, you will generate questions that can be asked by a user to retrieve information from a large documentary corpus.

**Requirements:**

1. The synthesized question must be a single, self-contained question and must not use "and" to connect multiple questions.

2. The answer of the synthesized question will only be found in the table (within ⟨table⟩ and ⟨/table⟩) and cannot be found in any sentences outside the ⟨table⟩ and ⟨/table⟩ in the chunks of the provided contexts.

3. The synthesized question must require chunks/contexts to locate the table and cannot mention the 'table' directly.

4. The synthesized question must be specific enough to locate the contexts in a large documentary corpus.

5. Do not ask "what XYZ in the table"; the question must be general enough to be asked in a large corpus.

6. If you cannot synthesize a question which can only be answered in the table based on the above requirements, do not synthesize anything.

7. Provide an explanation why the answer can only be found in the table and cannot be found in other parts of the chunks/contexts.

8. Emphasize reasoning, aggregation, temporal comparison, or retrieval from source data. Imagine the question being asked without the table still making partial sense.

**Question Template:**

- Use the following template to generate the QA:

    ```
    {{TEMPLATES}}
    ```

**Output Format:**

```
{
    "questions": [
        {
            "question": "<synthesized-question>",
            "answer": "<answer-of-the-question>",
            "question_type": <choose from "factual_retrieval", "comparison",
                "summarization", "causal_reasoning">,
            "image": "<<tab-aaaaa>>",
            "explanation-table": "<explanation-table>",
            "explanation-chunks": "<explanation-chunks>",
            "sentences-chunks-used":
            {"Chunk1": "sentences-chunk1",
                "Chunk2": "sentences-chunk2", ...}
        },...
    ]
}
```

**Input Data:**

- Contexts: "{{contexts}}"
- Table: The table is included as '⟨table⟩... ⟨/table⟩' in the context.

**Notes:**

- If the table can be used only for visualization or illustration, return an empty list for 'sentences-chunks-used'.
- If you cannot use all the chunks in the answer, return an empty list for 'sentences-chunks-used'.

---

## B.6 REWRITING PROMPTS

---

**Prompt P.5: Question Rewriting**

Prompt: You are tasked with rewriting the following question in two different ways, using only the provided Contexts and without hallucinating any information.
**Date** {{current_date}}
**Tasks:**

1. **Specific Rewrite**: Add or substitute minimal keywords to tie the question to the Contexts, making retrieval unique while preserving meaning.

2. **Obscured Rewrite**: Paraphrase the specific version to reduce keyword overlap while keeping all needed details intact.

**Requirements:**

• No hallucinated facts.

• Do not remove critical content.

• Avoid source-referencing phrases ("in figure", "in table", etc.).

• Rewrites must be standalone, fluent, faithful to Contexts.

• Only add essential keywords (avoid over-specification).

Check if the original answer remains fully correct for both rewrites. If not, set `"answer_wrong"` = `"True"`, else `"False"`.
**Output Format:**

```
{
  "specific_question":
  "More specific version with essential keywords.",
  "obscured_question":
  "Paraphrased version with reduced keyword overlap.",
  "answer_wrong": "True/False"
}
```

**Example 1:** Original: "What is the revenue growth shown in Figure 3 in 2024's report?"

```
{
  "specific_question":
  "What is the revenue growth for Company XYZ in 2024?",
  "obscured_question":
  "How did XYZ's financial outcomes change in 2024?",
  "answer_wrong": "False"
}
```

**Example 2:** Original: "What is the median differential rate between hurdle rates and costs of capital for cyclical and non-cyclical firms?"

```
{
  "specific_question":
  "What is the median differential between hurdle
  rates and costs of capital for cyclical vs. non-cyclical firms in
  the S&P 500 according to the Corporate Finance Advisory?",
  "obscured_question":
  "Within the Corporate Finance Advisory, what is the
  median gap between
  required returns and capital costs for S&P 500 firms
  sensitive to the economy vs. stable sectors?",
  "answer_wrong": "False"
}
```

## B.7 ANSWER REWRITING PROMPTS

---

**Prompt P.6: Answer Rewriting**

Prompt: You are tasked with rewriting the following answer so that it contains all the facts for answering the question, given the contexts and the image.

**Instruction:**

- Do not hallucinate any additional information. Use only the provided contexts and images.
- The rewritten answer must include the **old correct answer**, if it is correct.
- If the answer is already complete, you may leave it unchanged.
- Make the answer as concise as possible.
- If the **old correct answer** is incomplete, expand it so that the `"complete_answer"` fully addresses the question.

**Output Format:**

```
{
    "complete_answer": "Final rewritten answer that is concise,
    faithful to contexts and images, and fully answers the question."
}
```

**Input Data:**

- Question: "{{rewritten_question_obscured}}"
- Contexts: "{{contexts}}"
- Old Correct Answer: "{{answer}}"
- Images: The image is as follows:

---

# C  HUMAN ANNOTATION

Annotators were provided with the following instructions to evaluate the quality of synthesized questions and responses against source documents.

## C.1  TASK OVERVIEW

The primary task is to read a synthesized question and response, then evaluate their quality based on the provided PDF pages and images. The core evaluation criterion is factuality.

## C.2  FACTUALITY EVALUATION

Annotators must determine whether the question and response are factually supported by the source material.

### C.2.1  PROCEDURE

Annotators were instructed to follow these steps:

1. Open the folder corresponding to the given ID.
2. Read the text from the PDF pages located in the chunk_X subfolder. Annotators were told to read all text, including tables and image captions, but to ignore the content of the images themselves.
3. Review the images in the img_X subfolder to understand which image is being referenced, then locate that image within the source PDF to read its context and caption.
4. Read the provided Question and Response pair.
5. Assign a factuality label to both the question and the response.

### C.2.2  LABEL DEFINITIONS

**Factuality-Question: Factual** All facts and claims in the question are directly supported by the source material. There are no hallucinations or unsupported statements.

**Factuality-Question: Not Factual** One or more facts or claims in the question are not supported by the source (i.e., contain hallucinated or fabricated content).

**Factuality-Response: Factual** All facts and claims in the response are directly supported by the source material. There are no hallucinations or unsupported statements.

**Factuality-Response: Not Factual** One or more facts or claims in the response are not supported by the source (i.e., contain hallucinated or fabricated content).

**Note:** The original instructions included a rule stating, "If a question or response is not factual, it should be labeled as 'Incomplete'." However, the provided examples use the "Not Factual" label, which was the standard followed during annotation.

### C.2.3  EXAMPLES

The following examples were provided to the annotators for guidance.

```
{
  "id": 0,
  "question": "What is the logo of a major telecommunications company
   mentioned in the context related to personalization strategies?",
  "response": "AT&T",
}

# Steps:
# 1. I open folder "0", read all the chunks and images.
# 2. The question seems factual from one of the chunk.
# 3. The response seems to NOT be the correct answer.
```

```
# Then, I label Factual-Question as `Factual`
# Then, I label Factual-Response as `Not Factual`
```

Listing 1: Example of a factual question with a non-factual response.

```
{
  "id": 4,
  "question": "What businesses are located near the proposed development
    area in the Project Catalyst?",
  "response": "AT&T",
}

# Steps:
# 1. I open folder "4", read all the chunks and images.
# 2. The question seems to be NOT factual because I did not see Project
    Catalyst in the pdf or images.
# 3. The response seems to be incorrect because the question is not
    factual.

# Then, I label Factual-Question as `Not Factual`
# Then, I label Factual-Response as `Not Factual`
```

Listing 2: Example of a non-factual question and response.

### C.3   COMPLETENESS EVALUATION

This task assesses whether the response provides all the necessary information to fully answer the question, based on the provided source material.

#### C.3.1   PROCEDURE

The procedure for evaluating completeness is identical to the factuality task: annotators must review all provided PDF chunks and images before making a judgment.

#### C.3.2   LABEL DEFINITIONS

**Complete:** The response includes all the required facts and details present in the source material needed to comprehensively answer the question.

**Incomplete:** The response omits one or more facts or claims that are present in the source and are necessary to fully answer the question.

EXAMPLE 1: INCOMPLETE RESPONSE

```
{
  "id": 2,
  "question": "What businesses are located near the proposed development
    area in the Project Catalyst?",
  "response": "AutoZone Auto Parts, Pizza Hut, Sonic Drive In, Joe's
    Pizza Italian",
}

# Steps:
# 1. I open folder "2", read all the chunks and images.
# 2. The response seems to miss: "Mr Jim's Pizza, Justin Spirits, Allsup'
    s Convenience Store."

# Then, I label Completeness as `Incomplete`
```

Listing 3: Example of a response that is missing information available in the source document.

EXAMPLE 2: COMPLETE RESPONSE

```
{
  "id": 0,
  "question": "What is the logo of a major telecommunications company
    mentioned in the context related to personalization strategies?",
  "response": "AT&T",
}

# Steps:
# 1. I open folder "0", read all the chunks and images.
# 2. The response seems to be complete. AT&T is the only answer.

# Then, I label Completeness as 'Complete'
```
Listing 4: Example of a response that contains all necessary information.

## C.4  GROUNDING VERIFICATION

For each question, annotators were required to verify which specific source materials (PDF text chunks or images) were necessary to formulate the answer.

### C.4.1  PROCEDURE AND LABEL DEFINITIONS

**Grounding Verification-chunk-X:**  After reading the question, the annotator must determine if the text content of chunk_X.pdf contains any information used in, or required for, the answer.

- **Required:** The chunk's text contains information needed to answer the question.
- **Not Required:** The chunk's text does not contain any relevant information.

**Grounding Verification-img-X:**  The annotator must determine if img_X (including its caption and context within the PDF) contains any information used in, or required for, the answer.

- **Required:** The image or its caption contains information needed to answer the question.
- **Not Required:** The image and its caption do not contain any relevant information.

EXAMPLE: GROUNDING VERIFICATION

```
{
  "id": 0,
  "question": "What businesses are located near the proposed development
    area in the Project Catalyst?",
  "response": "AutoZone Auto Parts, Pizza Hut, Sonic Drive In, Joe's
    Pizza Italian",
}

# Steps for chunk-0:
# 1. I open folder "0" and then the sub-folder chunk_0.
# 2. I read the text within pages.pdf.
# 3. I find part of the answer to the question in the text.
# 4. I label 'Grounding Verification-chunk-0' as 'Required'.

# Steps for chunk-1:
# 1. I check for a sub-folder named chunk_1 in folder "0".
# 2. No chunk_1 sub-folder exists, so I skip this label.

# Steps for img-0:
# 1. I open folder "0" and then the sub-folder img_0.
# 2. I view img_0.jpg and locate it in the original PDF to check its
    context.
# 3. I find part of the answer to the question in the image.
# 4. I label 'Grounding Verification-img-0' as 'Required'.
```

```
# Steps for img-1:
# 1. I open folder "0" and then the sub-folder img_1.
# 2. I view img_1.jpg and its context.
# 3. I do NOT find any part of the answer in this image.
# 4. I label 'Grounding Verification-img-1' as 'Not Required'.
```

Listing 5: Example demonstrating how to label individual source chunks and images as required or not required.

## C.5 Self-Contained Evaluation

This task assesses whether a question is understandable and complete on its own, without needing external context or references to specific, unnamed documents.

### C.5.1 Procedure

Annotators were instructed to read only the question and determine if it could be understood and answered without ambiguity, assuming one had access to a large database of documents.

### C.5.2 Label Definitions

**True:** The question is self-contained. It is clearly phrased, makes sense on its own, and provides enough specific detail (e.g., names, topics, concepts) to be answerable. It does not rely on vague document references. For example, "What are the key benefits of solar energy mentioned in the 2022 Department of Energy report?" is self-contained.

**False:** The question depends on external or implicit context to be meaningful. It may contain vague deictic references (e.g., "in the image above," "according to this chart," "what does this mean?") without clarifying what the reference points to. For example, "What is the logo in the image?" is not self-contained as it requires seeing a specific, un-referenced image.

EXAMPLE 1: NOT SELF-CONTAINED

```
{
  "id": 1,
  "question": "What is the logo in the image?",
  "response": "AT&T",
}

# Steps:
# 1. I read the question.
# 2. I find it is NOT clear; "what image?" is an unanswered prerequisite.
# 3. I label 'Self-Contained' as 'False'.
```

Listing 6: Example of a question that is not self-contained due to a vague reference ("the image").

EXAMPLE 2: SELF-CONTAINED

```
{
  "id": 0,
  "question": "What is the logo of a major telecommunications company
    mentioned in the context related to personalization strategies?",
  "response": "AT&T",
}

# Steps:
# 1. I read the question.
# 2. I find it is clear. I can use the information within the question to
     search for a relevant document.
# 3. I label 'Self-Contained' as 'True'.
```

Listing 7: Example of a question that is self-contained because it provides sufficient context ("personalization strategies," "telecommunications company").

## C.6 HUMAN-LIKE INTENT EVALUATION

This task assesses whether a question reflects a natural and meaningful information-seeking intent, typical of a human user interacting with a document or database.

### C.6.1 PROCEDURE

Annotators were instructed to read the question and judge its authenticity as a genuine human query. The focus was on the nature of the question's intent rather than its grammatical perfection.

### C.6.2 LABEL DEFINITIONS

**True:** The question represents a reasonable and natural query a human would make. It seeks meaningful information such as facts, summaries, comparisons, or explanations, and is phrased in a way that reflects a real information need. For example: "What were the company's main revenue streams in the last fiscal year?"

**False:** The question is unnatural, trivial, or does not reflect a plausible human intent. This includes questions that are overly literal (e.g., counting word occurrences), focus on formatting (e.g., font sizes), are phrased robotically, or seek bizarrely specific details that a human would be unlikely to ask.

EXAMPLE 1: NOT HUMAN-LIKE

```
{
  "id": 1,
  "question": "How many logos in the Figure one of the major
    telecommunications company?",
  "response": "13",
}

# Steps:
# 1. I read the question.
# 2. I do not think a person using an information retrieval system would
    ask this style of question.
# 3. I label 'Human-like' as 'False'.
```

Listing 8: Example of a question that is not human-like due to its trivial, count-based nature.

EXAMPLE 2: HUMAN-LIKE

```
{
  "id": 3,
  "question": "What were the top two revenues for the EMS division in
    2012?",
  "response": "In 2012, the revenues were approximately HK$493,208,000
    and HK$391,677,000.",
}

# Steps:
# 1. I read the question.
# 2. I find it is clear and reflects a specific, meaningful financial
    inquiry.
# 3. I label 'Human-like' as 'True'.
```

Listing 9: Example of a question that reflects a clear, natural, and meaningful information need.

# D  EXAMPLES

## D.1  EXAMPLES FOR TEXT-RETRIEVAL BETTER THAN IMAGE-RETRIEVAL

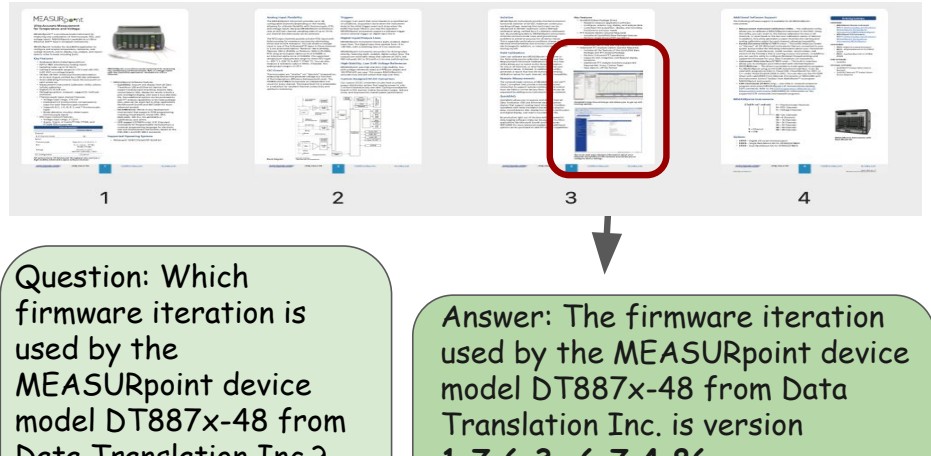

**Question:** Which firmware iteration is used by the MEASURpoint device model DT887x-48 from Data Translation Inc.?

**Answer:** The firmware iteration used by the MEASURpoint device model DT887x-48 from Data Translation Inc. is version **1.7.6.3, 6.7.4.86.**

**Text:** The firmware iteration used by the MEASURpoint device model DT887x-48 from Data Translation Inc. is **1.7.6.3, 6.7.4.86.** ✅

**Img:** The firmware iteration used by the MEASURpoint device model DT887x-48 from Data Translation Inc. is not explicitly stated in the provided images. The documents mention software and calibration utilities but do not specify a firmware version for the DT887x-48. ❌

**GME:** The firmware iteration used by the MEASURpoint device model DT887x-48 from Data Translation Inc. is displayed by the Eureka Discovery Utility, which provides the version of the firmware running on your instrument. ❌

**T+I:** The MEASURpoint device model DT887x-48 from Data Translation Inc. uses firmware version 1.7.6.3, 6.7.4.86. ✅

Figure 4: Image-retrieval system fails to extract factual facts and details.

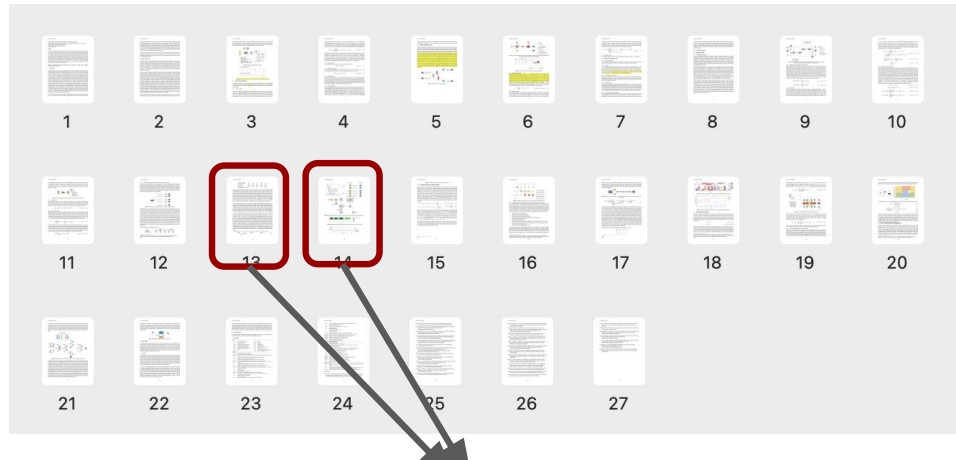

Question: How many MIbs are produced daily on average for Grade 1 over the span of weeks 0 to 55?

Answer: The average daily production level for Grade 1 varies between 179 and 243 MIbs over the span of weeks 0 to 55.

**Text**: The average daily production for Grade 1 over weeks 0 to 55 ranges from **179 MIbs to 243 MIbs**.

**Img**: The average daily production for Grade 1 over the span of weeks 0 to 55 is approximately **2,000 MIbs** (as indicated by the blue "Avg Daily Production (MIbs)" line in Figure 10). 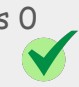

**GME**: Based on Figure 10, the average daily production for Grade 1 over weeks 0 to 55 is approximately **140 MIbs**.

**T+I**: The average daily production for Grade 1 over weeks 0 to 55 ranges from **179 to 243 MIbs**. 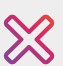

Figure 5: Image-retrieval system fails to extract factual facts and details in the image.

## D.2 EXAMPLES FOR IMAGE-RETRIEVAL BETTER THAN IMAGE-RETRIEVAL

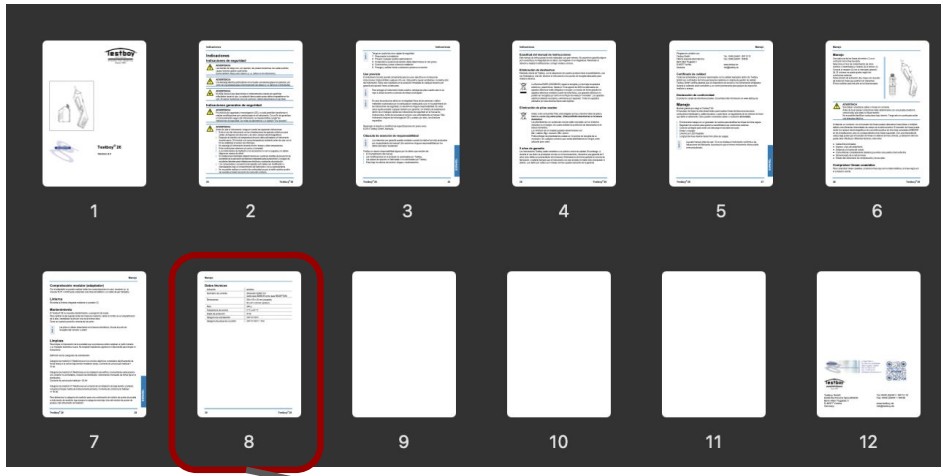

**Question:** Can you provide the service temperature interval for the Testboy® 26?

**Answer:** The service temperature interval for the Testboy® 26 is 5 °C to 40 °C.

**Text**: The provided contexts do not contain any information about the Testboy® 26 or its service temperature interval. Therefore, I cannot provide the service temperature interval for the Testboy® 26 based on the listed data. ❌

**Img**: The service temperature interval for the Testboy® 26 is 5 °C to 40 °C. ✅

**GME**: The service temperature interval for the Testboy® 26 is 5 °C to 40 °C. ✅

**T+I**: The service temperature interval for the Testboy® 26 is 5 °C to 40 °C. ✅

Figure 6: Text-retrieval system fails to extract factual facts and details in the table.

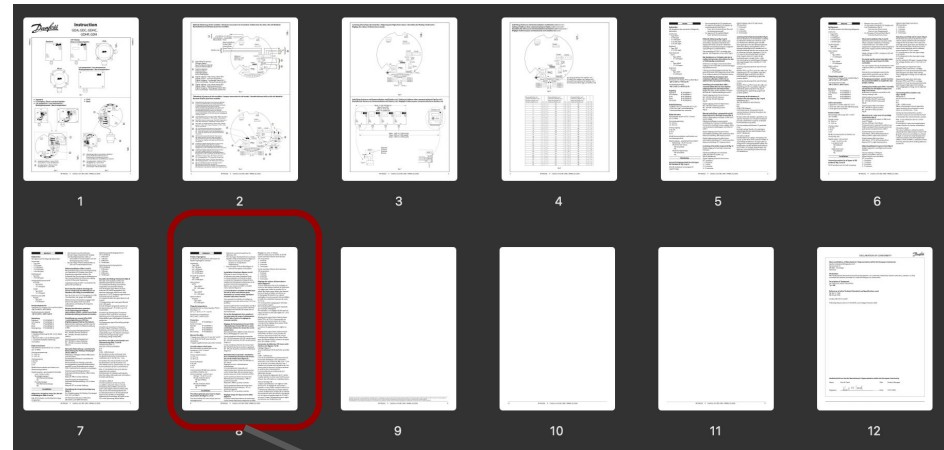

Question: How are the IP protection levels classified for different detector models, such as standard and LCD, used in tough conditions?

Answer: Standard and Écran à cristaux liquides detectors have an IP protection level of IP 30 (NEMA 1). Models with IP 65 and EExd have IP 65 (NEMA 4), and the low-temperature model has IP 40 (NEMA 2).

**Text**: The IP protection levels for different detector models, such as standard and LCD, are classified as follows: - Standard and LCD models: No specific IP rating mentioned, but grounding is required for safety. - IP65 models: Protected against dust and low-pressure water jets from any direction (IP65 rating). ✗

**Img**: The IP protection levels for different detector models are classified as follows: - **Standard and LCD display models:** IP 30 (NEMA 1) - **IP 65 model:** IP 65 (NEMA 4) - **EExd model:** IP 65 (NEMA 4) - **Low temperature model:** IP 40 (NEMA 2) ✓

**GME**: The IP protection levels for different detector models are as follows: - Standard and LCD display models: IP 30 (NEMA 1) - IP 65 and EExd models: IP 65 (NEMA 4) - Low temperature model: IP 40 (NEMA 2) For tough conditions, IP 65 and EExd models are used, providing high protection against dust and water jets. ✓

**T+I**: The IP protection levels for different detector models are classified as follows: - Standard and LCD display: IP30 (NEMA 1) - IP 65 model: IP65 (NEMA 4) - EExd model: IP65 (NEMA 4) - Low temperature model: IP40 (NEMA 2) ✓

Figure 7: Text-retrieval system fails to extract factual facts and details in the table.

### D.3 EXAMPLES FOR MULTIMODAL-RETRIEVAL BETTER THAN SINGLE-MODALITY-RETRIEVAL

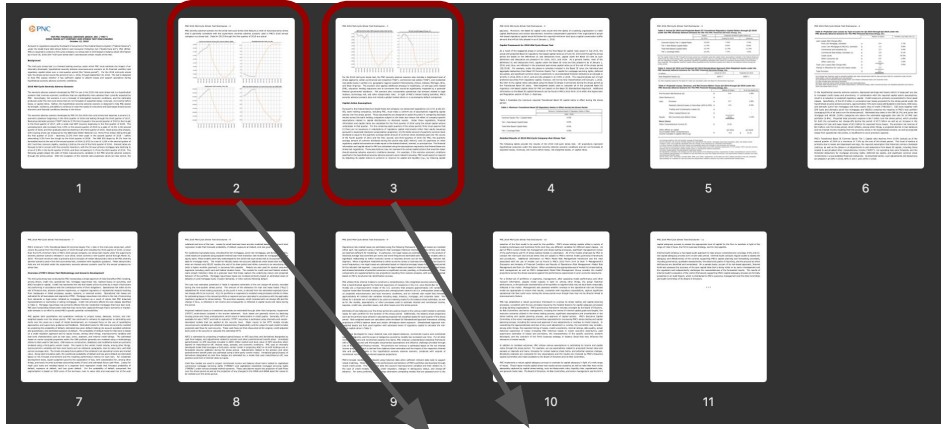

**Question:** According to the PNC's 2016 stress evaluation, how did nonfarm employment levels change between mid-2015 and the third quarter of 2018 in their severe case?

**Answer:** The total nonfarm employment decreased from 144,000 in 2015 Q2 to around 136,000 in 2017 Q1 and then started to recover, reaching approximately 137,000 by 2018 Q3.

**Text**: According to PNC's 2016 stress evaluation, nonfarm employment levels declined from about 144,000 in mid-2015 to around 136,000 by the third quarter of 2018 in their severe case. ❌

**Img**: According to PNC's 2016 stress evaluation, nonfarm employment levels declined sharply from about 142 million in mid-2015 to a low of around 136 million by the third quarter of 2018 in their severe case. ❌

**GME**: According to PNC's 2016 stress evaluation, nonfarm employment levels declined from about 142,000 thousand in mid-2015 to approximately 137,000 thousand by the third quarter of 2018 in their severely adverse scenario. 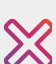

**T+I**: According to PNC's 2016 stress evaluation, nonfarm employment levels declined from about 144,000 (thousands) in mid-2015 to a low near 136,000 (thousands) before partially recovering to around 137,000 (thousands) by the third quarter of 2018 in their severe case. 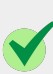

Figure 8: MM RAG system handles multi-modality-evidence questions better.

Table 7: Retrieval performance (`Precision@10` / `Recall@10`) of four RAG systems on 1600 QA pairs, averaged across eight domains and broken down by question and answer types.

| Type | Text (OpenAI) | | IMG (`colqwen`) | | MM (GME) | | T+I | |
|---|---|---|---|---|---|---|---|---|
| | *Prec.* | *Recall* | *Prec.* | *Recall* | *Prec.* | *Recall* | *Prec.* | *Recall* |
| Factual Retrieval | 0.319 | 0.759 | 0.237 | 0.839 | 0.304 | 0.876 | 0.416 | 0.862 |
| Comparison | 0.440 | 0.839 | 0.276 | 0.854 | 0.368 | 0.901 | 0.503 | 0.891 |
| Summary | 0.497 | 0.856 | 0.329 | 0.830 | 0.400 | 0.907 | 0.563 | 0.883 |
| Logical | 0.496 | 0.801 | 0.306 | 0.789 | 0.381 | 0.832 | 0.537 | 0.829 |
| Text-only | 0.511 | 0.821 | 0.324 | 0.774 | 0.390 | 0.836 | 0.558 | 0.820 |
| Img-only | 0.152 | 0.751 | 0.174 | 0.922 | 0.258 | 0.900 | 0.273 | 0.916 |
| Text + Img | 0.489 | 0.850 | 0.306 | 0.833 | 0.392 | 0.907 | 0.555 | 0.880 |
| Table-required | 0.431 | 0.773 | 0.270 | 0.798 | 0.339 | 0.872 | 0.493 | 0.851 |

Table 8: Retrieval performance (Precision@10 / Recall@10) and end-to-end performance (`Recall` using retrieved-top-10 and retrieved-top-20 candidates) of two MM-RAG systems on 200 QA pairs across eight domains, with average recall reported across all domains.

| Domain | MM (`Voyage`) | | | | MM (GME) | | | |
|---|---|---|---|---|---|---|---|---|
| | Retrieval | | End-to-end | | Retrieval | | End-to-end | |
| | Prec. | Recall | *top*-10 | *top*-20 | Prec. | Recall | *top*-10 | *top*-20 |
| Commerce | 0.518 | 0.892 | 0.629 | 0.653 | 0.354 | 0.895 | 0.617 | 0.611 |
| Construction | 0.406 | 0.733 | 0.603 | 0.609 | 0.336 | 0.881 | 0.601 | 0.616 |
| CRM | 0.418 | 0.748 | 0.634 | 0.653 | 0.343 | 0.884 | 0.623 | 0.637 |
| Education | 0.419 | 0.784 | 0.652 | 0.658 | 0.366 | 0.912 | 0.640 | 0.668 |
| Energy | 0.418 | 0.783 | 0.659 | 0.680 | 0.331 | 0.847 | 0.669 | 0.666 |
| Finance | 0.426 | 0.726 | 0.622 | 0.644 | 0.370 | 0.898 | 0.627 | 0.636 |
| Healthcare | 0.388 | 0.766 | 0.638 | 0.668 | 0.376 | 0.857 | 0.642 | 0.664 |
| Legal | 0.431 | 0.764 | 0.631 | 0.669 | 0.327 | 0.876 | 0.609 | 0.629 |
| Avg. | 0.416 | 0.777 | 0.633 | 0.654 | 0.350 | 0.881 | 0.628 | 0.641 |

# E    ADDITIONAL EXPERIMENTS

## E.1    RETRIEVAL PERFORMANCE

We break down retrieval performance by question and answer types, as reported in Table 7. We find that question type has minimal impact on retrieval recall, whereas answer type plays a significant role. For text-only retrieval, performance is substantially higher on questions requiring text to answer, but markedly lower on image-required questions. Conversely, for image-only retrieval, questions requiring image-based answers are retrieved more effectively than those requiring text, highlighting the modality-specific strengths of each embedding approach. Combining both embeddings (T+I) effectively leverages the advantages of each modality, resulting in higher overall recall. For multimodal embeddings, image-required questions tend to be retrieved more easily than text-required questions, suggesting that current multimodal embeddings function more like image retrieval in practice.

## E.2    MM-EMBEDDING RAG COMPARISON

We compare RAG performance using two multimodal embeddings: `voyage-multimodal-3` and `gme-Qwen2-VL-7B-Instruct`, with results reported in Table 8 and Table 9. While `voyage-multimodal-3` achieves slightly lower recall but higher precision in retrieval compared to `gme-Qwen2-VL-7B-Instruct`, it delivers better overall performance when integrated into MM-RAG.

## E.3    COST COMPARISON

We also calculate the average inference cost and latency of different RAG systems. The image-only system (IMG) is the most efficient, while multimodal systems (MM) are the slowest, reflecting

Table 9: Precision and recall of two MM-RAG systems using the top 10 retrieved chunks retrieved by their retrievers, evaluated across different question and answer types on $1,600$ QA pairs spanning eight domains, with average recall reported across all domains.

| Type | MM (Voyage) | | MM (GME) | |
|---|---|---|---|---|
| | Prec. | Recall | Prec. | Recall |
| Factual Retrieval | 0.606 | 0.595 | 0.691 | 0.580 |
| Comparison | 0.656 | 0.604 | 0.730 | 0.608 |
| Summary | 0.694 | 0.738 | 0.802 | 0.655 |
| Logical Reasoning | 0.699 | 0.727 | 0.837 | 0.679 |
| Text-only | 0.871 | 0.824 | 0.868 | 0.759 |
| Img-only | 0.414 | 0.348 | 0.436 | 0.312 |
| Text+Img | 0.786 | 0.656 | 0.810 | 0.636 |
| Table-required | 0.832 | 0.736 | 0.867 | 0.750 |

Table 10: Average cost of different RAG systems.

| | IMG | TEXT | MM (GME) | MM (T+I) |
|---|---|---|---|---|
| Avg. Cost ($) | 0.012 | 0.036 | 0.022 | 0.029 |
| Avg. Latency (s) | 5.606 | 7.290 | 7.897 | 9.383 |

the trade-off between complexity and capability. The text-only system consumes the most tokens and is therefore the most expensive. The T+I fusion RAG retrieves from text chunks first, then images, which increases latency. These results suggest that modern MM-RAG systems can offer both improved performance and lower cost compared to text-only RAG.

# F   ADDITIONAL ANALYSIS

## F.1   CONTENT-RICH IMAGES INCREASE DIFFICULTY

We analyzed all images in the documents of the easiest domain (*commerce manufacturing* and *legal*) and the most difficult domains (*finance* and *construction*). Using `gemini-2.5-pro`, we classified images as either content-rich (providing information not present in the text) or illustrative. In finance and construction, 62.8% and 69.3% of images, respectively, were content-rich, compared to 40.0% in commerce manufacturing and 49.5% in legal. This suggests that domains with a higher proportion of content-rich images present a greater challenge for RAG, as these images require effective multimodal understanding beyond text.

## F.2   QUESTION TYPE AFFECTS DIFFICULTY

As shown in Section 4.2, the type of context required to answer a question is the most significant factor influencing RAG performance. Different categories of questions contribute unevenly to the advantage of either text- or image-retrieval RAG systems. By carefully analyzing questions that can only be answered correctly by one of the two systems, we summarize the key distinguishing features:

Text-Retrieval Advantages:

- *Entity Recognition* (e.g., brands, organizations; 53.9% of text advantage): Strong at identifying specific people, companies, or organizations.
- *Comparative Analysis* (37.6%): Ranking, evaluating differences, or determining which option is preferable.
- *Contextual Numerical Reasoning* (34.8%): Numbers requiring understanding of surrounding context.
- *Quantity Estimation* (29.1%): Questions asking about amounts, counts, or measurements.
- *Domain-Specific Terminology* (16.3%): Technical, scientific, or specialized terms and standards.

Image-Retrieval Advantages:

- *Visual Chart Data Interpretation* (64.2% of image wins): Charts and tables make numerical information more accessible. *Example:* How much of the auto ABS senior tranches in Europe were rated AAA in early 2018?
- *Temporal / Chronological Data* (40.0%): Timeline visualizations clarify temporal relationships. *Example:* When did U.S. petroleum imports drop under $20 billion?
- *Technical / Measurement Information* (19.2%): Diagrams often contain measurements or specifications not in text. *Example:* What is the service temperature interval for Testboy® 26 based on the listed data?
- *Spatial / Geographic Reasoning* (13.3%): Maps and layouts convey location context and spatial relationships. *Example:* What is the impact of delivery time on scheduling at 22 Bishopsgate?

## F.3   DOCUMENT FORMATS DO NOT AFFECT PERFORMANCE.

As discussed in Section 3.1, documents span formats such as newspapers, textbooks, webpages, forms, reports, papers, slides, and posters. In the best-performing domain, *commerce manufacturing*, the distribution is diverse, with reports (45.2%), textbooks (23.6%), papers (18.7%), and webpages (10.5%). In contrast, the worst-performing domain, *finance*, is dominated by reports (80.8%), with only small shares of papers (12.2%), textbooks (2.9%), and webpages (2.3%). Yet this trend is not consistent: the second-worst domain, *construction*, is also diverse, with reports (53.9%), papers (30.4%), and textbooks (11.3%). Therefore, format distribution alone cannot explain performance differences.

**Document layouts do not affect performance.** In the best-performing domain, *commerce manufacturing*, documents are composed of text (73.9%), tables (4.0%), and figures (22.1%), while the worst-performing domain, *finance*, shows a nearly identical distribution (72.9% text, 3.7% tables, 23.4% figures). Since all domains exhibit similar layout patterns, layout does not appear to be a key factor in RAG performance.

### F.4 DOCUMENT PAGE NUMBERS DO NOT AFFECT PERFORMANCE.

In the best-performing domains (*commerce manufacturing*, *education*, and *legal*), the average lengths are 13.1, 14.6, and 12.6 pages, respectively. In contrast, the worst-performing domains (*finance*, *construction*, and *healthcare*) average 15.4, 12.9, and 12.1 pages. These small differences suggest that document length is not a major factor in RAG performance.

