# OpenReview forum: "UniDoc-Bench: A Unified Benchmark for Document-Centric Multimodal RAG"
_ICLR.cc/2026/Conference — ICLR 2026 Conference Withdrawn Submission_

### Official Review · Reviewer_7MPR · 2025-10-26

**Soundness:** 3
**Presentation:** 3
**Contribution:** 2
**Rating:** 4
**Confidence:** 3

**Summary:**

This introduce UniDoc-Bench, a large-scale, realistic benchmark for MM-RAG built from 70k real-world PDF pages across 8 domains. It supports 4 different types of retrieval paradigms: text-only, image-only,  multimodal text–image fusion and multimodal joint retrieval. The paper further offers analysis comparing visual and textual context usage in sota multimodal LLMs.

**Strengths:**

1. The paper introduce a novel benchmark UniDoc-Bench which contains 1, 600 multimodal QA pairs spanning factual retrieval, comparison, summarization, and logical reasoning queries.
2. The paper introduce a fair and reproducible evaluation framework by fixing candidate pools across modalities , and measuring retrieval effectiveness, answer faithfulness, and completeness end-to-end across different RAG systems.
3. This paper conducts a systematic comparison of text-retrieval, image-retrieval, text–image fusion, multimodal joint retrieval pipelines, analyzing which retrieval strategy performs best under different question types, evidence modalities.

**Weaknesses:**

1. The significant contribution that I take away from this paper vs the previous work is unified evaluation and multiple reference. Therefore,
What are the benefits of unified evaluation?
2. How is multiple reference used  during evaluation to enhance the groundness?

**Questions:**

1. See the in weakness.
2. Will this paper opensource their evaluation framework?

---

> ### Author Response · Authors · 2025-11-18
> **Unified Protocol Ensures Unbiased Benchmarking**
>
> Reviewer: _"The significant contribution that I take away from this paper vs the previous work is unified evaluation and multiple reference. Therefore, What are the benefits of unified evaluation?"_
>
> **Our Response:**
>
> Thank you for highlighting this. The main benefit of our **unified evaluation** is that it enables a truly fair, “apples-to-apples” comparison across systems. Many prior works make claims like “image retrieval is all you need” or “multimodal retrieval is inherently better,” but these claims are often based on **unfair or inconsistent evaluation settings** --- for example, comparing multimodal models to weak text-only baselines that ignore non-text modalities during generation.
>
> Our unified protocol --- using standardized candidate pools, prompts, and evaluation metrics --- prevents such misleading conclusions by ensuring that **all models are evaluated under the same conditions**. We also create strong, fair baselines, such as text-only retrieval systems enhanced with captioned images, so comparisons are meaningful.
>
> This fair evaluation provides an unbiased view of what truly works. For example, we find that a simple text+image (T+I) RAG consistently outperforms joint multimodal embedding (MM) RAG. Surprisingly, even a strong text-only RAG baseline can outperform joint MM-RAG. These insights would be impossible to see under inconsistent evaluation setups and provide clear guidance for the field.

---

> ### Author Response · Authors · 2025-11-18
> **Using Multiple References to Enhance Grounding**
>
> Reviewer: _"How is multiple reference used during evaluation to enhance the groundness?"_
>
> **Our Response:**
>
> Thank you for the question. If by _“multiple reference”_ you mean leveraging multiple retrieved chunks to enhance grounding, our evaluation follows this approach. Specifically, we retrieve the top-K candidates (e.g., top-10 or top-20) and feed all of them to the generator, enabling the system to reference multiple evidence sources during response generation. We evaluate performance using metrics such as Recall and Precision. Please let us know if this does not align with your intended meaning.

---

> ### Author Response · Authors · 2025-11-18
> **Evaluation Framework Will Be Open-Sourced**
>
> Reviewer: _"Will this paper opensource their evaluation framework?"_
>
> **Our Response:**
>
> * Yes, we are committed to open-sourcing our work. The evaluation framework and associated code are already included in the supplementary materials (the .zip file) submitted with the paper.
> * We firmly believe in contributing reproducible research to the community and understand the importance of a practical, usable framework.
> * To that end, we plan to add more detailed documentation and guidance to help other researchers easily use our framework to evaluate their own RAG systems on our benchmark or their own custom data.

---

> ### Author Response · Authors · 2025-11-26
> **Friendly Reminder**
>
> We thank you for your constructive feedback. In response, we also have added **Long-Context VQA experiments**, as some reviewers requested clarification on the broader utility of our dataset. The new results show clear gains across domains: **GPT-5 achieves the highest Precision (0.658) and Claude Sonnet 4.5 achieves the highest Recall (0.708)**, outperforming Gemini 2.5 Pro, and Qwen2.5-VL-32B. These improvements demonstrate that our dataset is not only effective for standard RAG but also highly beneficial for long-context VQA scenarios. We kindly invite you to check the full VQA results in the comment in the main thread.
>
> We encourage you to take a look at the updated results and revisions. If these additions address your concerns, we would greatly appreciate a reconsideration of the scores, and we are happy to clarify any remaining questions. Thank you again for your time and helpful feedback.

---

### Official Review · Reviewer_7bma · 2025-10-30

**Soundness:** 3
**Presentation:** 3
**Contribution:** 2
**Rating:** 4
**Confidence:** 2

**Summary:**

This paper introduces UniDoc-Bench, a benchmark for document-centric multimodal retrieval-augmented generation (RAG). The dataset includes 1,600 synthetically generated multimodal QA pairs, of which 20% are validated by multiple annotators and expert adjudication to ensure quality. The benchmark standardizes evaluation across four paradigms: (1) text-only, (2) image-only, (3) multimodal text–image fusion, and (4) multimodal joint retrieval, enabling fair, apples-to-apples comparisons. Experimental results show that multimodal text–image fusion RAG systems perform best.

**Strengths:**

1. Provides a systematic and unified comparison of text retrieval, image retrieval, text–image fusion, and multimodal joint retrieval pipelines under consistent settings.
2. Ensures data reliability by validating 20% of the QA pairs through multiple annotators and expert adjudication.

**Weaknesses:**

The QA data are synthetically generated (GPT-4.1 and Gemini-Pro-2.5), which may introduce bias toward the LLMs used. This limits the benchmark’s ability to fully assess model generalization beyond this generation and templates distribution.

**Questions:**

Have you compared LLM-generated QA performance against fully human-written QA pairs to confirm generalizability?

---

> ### Author Response · Authors · 2025-11-18
> **Pipeline Design Reduces Model-Specific Bias**
>
> Reviewer: _The QA data are synthetically generated (GPT-4.1 and Gemini-Pro-2.5), which may introduce bias toward the LLMs used. This limits the benchmark’s ability to fully assess model generalization beyond this generation and templates distribution._
>
> **Our Response:**
>
> We appreciate this concern and have explicitly designed our pipeline to mitigate bias toward any single LLM.
>
> First, we **decouple generation and verification**: GPT-4.1 is used to generate QA candidates, while Gemini-Pro-2.5 is used for automatic checking, so no single model controls both steps.
>
> Second, the generation is **constrained by a large, human-authored template pool** (40–60 general templates per domain; see Appendix B.4), which provides diverse, human-designed query intents rather than free-form sampling from one model.
>
> Third, we conduct a **human-in-the-loop validation** stage, where three independent annotators and a senior expert adjudicator verify that questions are factual, self-contained, and exhibit realistic “human-like intent.”

---

> ### Author Response · Authors · 2025-11-18
> **Synthesized QA Closely Mimics Human Questions**
>
> Reviewer: _"Have you compared LLM-generated QA performance against fully human-written QA pairs to confirm generalizability?"_
>
> **Our Response:**
>
> We did not construct a separate test set written entirely by humans, because our goal is to build a **RAG data-synthesis pipeline** that can systematically generate high-quality QA pairs at scale. The generation process is strongly human-guided: it is grounded in a large pool of human-written templates and followed by multi-stage filtering. In addition, we invested ~640 annotator-hours to validate 20% of the QA pairs, using criteria such as **Human-like Intent**, for which 97.5% of sampled items passed (Tab 2). These results indicate that **our synthesized QA pairs closely mimic human-authored questions and support the benchmark’s generalizability**.

---

> ### Author Response · Authors · 2025-11-26
> **Friendly Reminder**
>
> We thank you for your constructive feedback. In response, we also have added **Long-Context VQA experiments**, as some reviewers requested clarification on the broader utility of our dataset. The new results show clear gains across domains: **GPT-5 achieves the highest Precision (0.658) and Claude Sonnet 4.5 achieves the highest Recall (0.708)**, outperforming Gemini 2.5 Pro, and Qwen2.5-VL-32B. These improvements demonstrate that our dataset is not only effective for standard RAG but also highly beneficial for long-context VQA scenarios. We kindly invite you to check the full VQA results in the comment in the main thread.
>
> We encourage you to take a look at the updated results and revisions. If these additions address your concerns, we would greatly appreciate a reconsideration of the scores, and we are happy to clarify any remaining questions. Thank you again for your time and helpful feedback.

---

### Official Review · Reviewer_wtt8 · 2025-10-31

**Soundness:** 3
**Presentation:** 3
**Contribution:** 3
**Rating:** 6
**Confidence:** 3

**Summary:**

The paper UniDoc-Bench introduces a large-scale benchmark for multimodal retrieval-augmented generation (RAG) across text, image, and table modalities, providing standardized evaluation for both open-source and commercial models.
It demonstrates that multimodal RAG improves performance and cost efficiency compared to text-only RAG, particularly in domains rich in visual content such as finance and construction.

**Strengths:**

1. Covers diverse modalities (text, image, multimodal, table-required) and question types, providing a unified evaluation framework.
2. Offers granular insights into the relative strengths of text and image retrieval and the role of domain-specific content richness.
3. Highlights cost and latency trade-offs in multimodal RAG, providing actionable insights for system optimization.

**Weaknesses:**

1. It would be beneficial to add more SoTA models' results such as Gemini and Claude with and without retrieved instances. Also, it'd be great to perform a more detailed comparison between different SoTA models.
2. Overrepresentation of finance and construction may bias generalization to less visually complex domains.

**Questions:**

How would different SoTA models (e.g., GPT, Gemini, Claude) perform on this benchmark and what types of errors do these models make?

---

> ### Author Response · Authors · 2025-11-18
> **Additional SoTA Models Evaluated for UniDoc-Bench (pt.1)**
>
> Reviewer: _"It would be beneficial to add more SoTA models' results such as Gemini and Claude with and without retrieved instances. Also, it'd be great to perform a more detailed comparison between different SoTA models."_
>
> **Our Response:** Thank you for the suggestion. We agree that adding results from more state-of-the-art models will strengthen our paper. Using (1) the best top-k values (10 and 20) from Table 3 and (2) the ground-truth images, tables, and text chunks (GT), we evaluated three additional advanced models: **GPT-5**, **Gemini-2.5-Pro**, and **Claude-4.5-Sonnet**.
>
> * Below are the **Faithfulness** results across different domains.
>
> | Domain | Gemini-2.5-Pro (topk=10) | Gemini-2.5-Pro (topk=20) | Gemini-2.5-Pro (GT) | Claude-4.5-Sonnet (topk=10) | Claude-4.5-Sonnet (topk=20) | Claude-4.5-Sonnet (GT) | GPT-5 (topk=10) | GPT-5 (topk=20) | GPT-5 (GT) |
> |--------|--------------------------|---------------------------|---------------------|----------------------------|-----------------------------|------------------------|-----------------|-----------------|------------|
> | Commerce Manufacturing | 0.7327 | 0.7218 | 0.7793 | 0.6844 | 0.6853 | 0.7845 | **0.7624** | **0.7689** | **0.8046** |
> | Construction | 0.6218 | 0.6289 | 0.7146 | 0.6259 | 0.5987 | 0.7154 | **0.6738** | **0.6954** | **0.7251** |
> | Crm | 0.6366 | 0.6644 | 0.7654 | 0.6299 | 0.6506 | 0.7575 | **0.6893** | **0.6968** | **0.8093** |
> | Education | 0.6798 | 0.6796 | 0.7698 | 0.5947 | 0.6004 | 0.7577 | **0.6863** | **0.7192** | **0.7964** |
> | Energy | 0.6973 | 0.7034 | **0.7819** | 0.6873 | 0.6772 | 0.7495 | **0.7353** | **0.7038** | 0.7515 |
> | Finance | 0.6385 | 0.6255 | 0.7161 | 0.6053 | 0.6034 | 0.6990 | **0.6826** | **0.6778** | **0.7334** |
> | Healthcare | 0.6531 | 0.6857 | 0.7594 | 0.5760 | 0.5955 | 0.7272 | **0.6898** | **0.7035** | **0.7694** |
> | Legal | 0.7342 | 0.7289 | 0.7757 | 0.6849 | 0.6698 | 0.7573 | **0.7553** | **0.7369** | **0.8174** |
> | **Average** | 0.6742 | 0.6798 | 0.7578 | 0.6361 | 0.6351 | 0.7435 | **0.7094** | **0.7128** | **0.7759** |
>
>
>
> * Below are the **Completeness** results across different domains.
>
> | Domain | Gemini-2.5-Pro (topk=10) | Gemini-2.5-Pro (topk=20) | Gemini-2.5-Pro (GT) | Claude-4.5-Sonnet (topk=10) | Claude-4.5-Sonnet (topk=20) | Claude-4.5-Sonnet (GT) | GPT-5 (topk=10) | GPT-5 (topk=20) | GPT-5 (GT) |
> |--------|--------------------------|---------------------------|---------------------|----------------------------|-----------------------------|------------------------|-----------------|-----------------|------------|
> | Commerce Manufacturing | 0.6871 | 0.7130 | 0.7005 | **0.8040** | **0.7977** | **0.8246** | 0.7229 | 0.7369 | 0.7692 |
> | Construction | 0.6481 | 0.6380 | 0.6455 | **0.6977** | **0.7016** | **0.7284** | 0.6407 | 0.6367 | 0.6946 |
> | Crm | 0.6586 | 0.7012 | 0.7511 | **0.7570** | **0.7603** | **0.8157** | 0.6603 | 0.6671 | 0.7707 |
> | Education | 0.6703 | 0.7131 | 0.7135 | **0.7366** | **0.7817** | **0.8196** | 0.6369 | 0.6947 | 0.7651 |
> | Energy | 0.6573 | 0.6801 | 0.6332 | **0.8108** | **0.7777** | **0.7978** | 0.7017 | 0.7048 | 0.7098 |
> | Finance | 0.6738 | 0.6659 | 0.6722 | **0.7753** | **0.7550** | **0.7721** | 0.7079 | 0.7021 | 0.7212 |
> | Healthcare | 0.6884 | 0.6754 | 0.7494 | **0.7762** | **0.7525** | **0.8459** | 0.6699 | 0.6851 | 0.7582 |
> | Legal | 0.7299 | 0.7327 | 0.6949 | **0.7673** | **0.7489** | **0.7902** | 0.7049 | 0.7297 | 0.7554 |
> | **Average** | 0.6767 | 0.6899 | 0.6950 | **0.7656** | **0.7594** | **0.7993** | 0.6806 | 0.6946 | 0.7430 |

---

> > ### Author Response · Authors · 2025-11-18
> > **Additional SoTA Models Evaluated for UniDoc-Bench (pt.2)**
> >
> > * Below are the **Faithfulness** results across different question and answer types.
> >
> > | Category | Gemini-2.5-Pro (topk=10) | Gemini-2.5-Pro (topk=20) | Gemini-2.5-Pro (GT) | Claude-4.5-Sonnet (topk=10) | Claude-4.5-Sonnet (topk=20) | Claude-4.5-Sonnet (GT) | GPT-5 (topk=10) | GPT-5 (topk=20) | GPT-5 (GT) |
> > |----------|--------------------------|---------------------------|---------------------|----------------------------|-----------------------------|------------------------|-----------------|-----------------|------------|
> > | **Image Only** | 0.4642 | 0.4724 | 0.5494 | 0.4344 | 0.4361 | 0.5155 | **0.5344** | **0.5480** | **0.6036** |
> > | **Image Plus Text As Answer** | 0.6192 | 0.6152 | 0.6542 | 0.5689 | 0.5734 | 0.6714 | **0.6501** | **0.6419** | **0.7126** |
> > | **Table Required** | 0.7871 | 0.8038 | **0.8689** | 0.7732 | 0.7524 | 0.8577 | **0.8037** | **0.8210** | 0.8390 |
> > | **Text Only** | 0.8050 | 0.8103 | 0.9452 | 0.7489 | 0.7589 | 0.9212 | **0.8315** | **0.8339** | **0.9469** |
> > | **Causal Reasoning** | 0.6847 | 0.7011 | **0.8542** | 0.6232 | 0.6098 | 0.8227 | **0.7488** | **0.7236** | 0.8496 |
> > | **Comparison** | 0.6632 | 0.6609 | 0.6752 | 0.6130 | 0.6286 | 0.6622 | **0.7091** | **0.6968** | **0.7126** |
> > | **Factual Retrieval** | 0.6615 | 0.6700 | 0.7470 | 0.6378 | 0.6382 | 0.7309 | **0.6915** | **0.7120** | **0.7652** |
> > | **Summarization** | 0.6847 | 0.6903 | 0.8048 | 0.6503 | 0.6306 | 0.8207 | **0.6936** | **0.7205** | **0.8339** |
> > | **Average** | 0.6742 | 0.6798 | 0.7578 | 0.6361 | 0.6351 | 0.7435 | **0.7094** | **0.7128** | **0.7759** |
> >
> >
> > * Below are the **Completeness** results across different question and answer types.
> >
> > | Category | Gemini-2.5-Pro (topk=10) | Gemini-2.5-Pro (topk=20) | Gemini-2.5-Pro (GT) | Claude-4.5-Sonnet (topk=10) | Claude-4.5-Sonnet (topk=20) | Claude-4.5-Sonnet (GT) | GPT-5 (topk=10) | GPT-5 (topk=20) | GPT-5 (GT) |
> > |----------|--------------------------|---------------------------|---------------------|----------------------------|-----------------------------|------------------------|-----------------|-----------------|------------|
> > | **Image Only** | 0.5688 | 0.5815 | 0.6332 | **0.6252** | 0.6019 | 0.6769 | 0.5909 | **0.6020** | **0.6815** |
> > | **Image Plus Text As Answer** | 0.6075 | 0.6362 | 0.6857 | **0.7081** | **0.7170** | **0.8096** | 0.6013 | 0.5999 | 0.7002 |
> > | **Table Required** | 0.7673 | 0.7730 | 0.7123 | **0.8502** | **0.8542** | **0.8057** | 0.7465 | 0.7741 | 0.7530 |
> > | **Text Only** | 0.7612 | 0.7619 | 0.7605 | **0.8526** | **0.8469** | **0.8940** | 0.7657 | 0.7843 | 0.8380 |
> > | **Causal Reasoning** | 0.6650 | 0.6863 | 0.6832 | **0.7815** | **0.8043** | **0.8601** | 0.6925 | 0.6869 | 0.7562 |
> > | **Comparison** | 0.7058 | 0.7131 | 0.7195 | **0.7951** | **0.7836** | **0.8043** | 0.6906 | 0.6845 | 0.7396 |
> > | **Factual Retrieval** | 0.6693 | 0.6828 | 0.6999 | **0.7266** | **0.7190** | **0.7536** | 0.6703 | 0.6957 | 0.7288 |
> > | **Summarization** | 0.6603 | 0.6658 | 0.6722 | **0.7757** | **0.7673** | **0.8496** | 0.6529 | 0.6856 | 0.7795 |
> > | **Average** | 0.6767 | 0.6899 | 0.6950 | **0.7656** | **0.7594** | **0.7993** | 0.6806 | 0.6946 | 0.7430 |

---

> > > ### Author Response · Authors · 2025-11-18
> > > **Additional SoTA Models Evaluated for UniDoc-Bench (pt.3)**
> > >
> > > * Faithfulness
> > >     - **GPT-5**: **Faithfulness Champion**
> > >       - Highest average Faithfulness (**0.7759 GT**).
> > >       - Most resilient to imperfect retrieval, dropping only **0.0631** from GT to topk=20.
> > >       - Excels on **Text Only**, **Table Required**, and **Causal Reasoning** questions, demonstrating robust reasoning.
> > >
> > >     - **Claude-4.5-Sonnet**: **Faithfulness weaker and retrieval-sensitive**
> > >       - Drops **0.1084** from GT (0.7435) to topk=20 (0.6351).
> > >       - Highly faithful when given perfect data but struggles with retrieved context.
> > >
> > >     - **Gemini-2.5-Pro**: **Solid Faithfulness, but below GPT-5**
> > >       - Average Faithfulness of **0.7578 GT**.
> > >       - Minor improvements from increasing k; generally robust but not the best.
> > >
> > >     - **All models**: Increasing top-k from 10 to 20 shows negligible or slightly negative impact on Faithfulness.
> > >
> > > * Completeness
> > >     - **Claude-4.5-Sonnet**: **Completeness Champion**
> > >       - Highest Completeness score (**0.7993 GT**).
> > >       - Strong across most answer types.
> > >
> > >     - **GPT-5**: **Good Completeness, slightly behind Claude**
> > >       - Average Completeness **0.7430 GT**.
> > >       - Competitive performance, especially for text-heavy questions.
> > >
> > >     - **Gemini-2.5-Pro**: **Limited by model capability**
> > >       - GT score (0.6950) almost identical to topk=20 (0.6899).
> > >       - Providing perfect information does not improve completeness; bottleneck is the model itself.
> > >
> > >     - **All models**: Struggle with **image-based questions**. Example: GPT-5 only 0.6036 (GT) on "Image Only" vs >0.92 on "Text Only". Faithful, multimodal reasoning remains challenging.
> > >
> > > * Model Trade-offs
> > >     - **GPT-5**: **Best for Faithfulness**, strong second for Completeness.
> > >     - **Claude-4.5-Sonnet**: **Best for Completeness**, weaker in Faithfulness.
> > >     - **Gemini-2.5-Pro**: Strong Faithfulness runner-up, weakest Completeness.
> > >     - **Recommendation**: Choose a model based on task priority—**Faithfulness** vs **Completeness**.

---

> ### Author Response · Authors · 2025-11-18
> **No Overrepresentation: Domains Intentionally Balanced**
>
> Reviewer: _"Overrepresentation of finance and construction may bias generalization to less visually complex domains."_
>
> **Our Response:**
>
> We would like to clarify that no domains are overrepresented in UniDoc-Bench. The benchmark is explicitly balanced across 8 domains in terms of page count and QA pairs (each domain contains 200 QA pairs, for a total of 1,600). The page distribution is:
>
> | Domain | # Pages | Percentage |
> |-------------|---------|------------|
>  | Education | 10,144 | 14.1% |
>  | Healthcare | 11,796 | 16.4% |
>  | CRM | 7,181 | 10.0% |
>  | Legal | 7,624 | 10.6% |
>  | Finance | 8,331 | 11.6% |
>  | Energy | 8,452 | 11.8% |
>  | Construction| 8,677 | 12.1% |
>  | Commerce | 9,649 | 13.4% |
>
> You are correct that domains such as Finance and Construction are more visually challenging. Our analysis in Appendix F.1 confirms this: they have a higher proportion of “content-rich” images (62.8% and 69.3%, respectively) compared to domains like Commerce (40.0%) and Legal (49.5%). **However, this is a design choice rather than a bias.** By providing a balanced set of 8 domains that span a wide range of visual complexity, UniDoc-Bench enables more robust and comprehensive evaluation of MM-RAG systems.

---

> ### Author Response · Authors · 2025-11-18
> **Typical Failure Patterns of SoTA Models**
>
> Reviewer: _"How would different SoTA models (e.g., GPT, Gemini, Claude) perform on this benchmark and what types of errors do these models make?"_
>
> **Our response:**
>
> Thanks for your question. We have include the new experiments results in the first response. We also did a case study below.
> In our experiments, different SoTA models (e.g., GPT, Gemini, Claude) show comparable overall trends, and we do not observe a single model that uniformly dominates across all question types or answer modalities. Instead, we conduct a manual, fine-grained analysis of their responses to characterize typical failure modes and error patterns. Specifically, we summarize these representative error types below with the same system prompts for different models to ensure apple-to-apple comparison.
>
> * For Faithfulness, we check all the question-answer pairs which **GPT-5** fails to answer faithfully **with ground-truth text and image chunks**, we find the following patterns.
>     * **Factual Inaccuracies and Data Extraction Errors**: This is the most frequent pattern. In these cases, the baseline answer provides a specific piece of information (like a number, date, or name) that directly contradicts the ground truth, often due to misreading a chart or figure..
>         * Example: The baseline incorrectly extracts a firmware version as "1.7.0.3" while the GT is "1.7.6.3".
>         * Example: The baseline extracts a completely incorrect firmware version ("8.00.00.00") and IP address ("169.254.103.111"), while the GT provides different values.
>     * **False Negatives (Refusals)**: The baseline incorrectly states that the requested information does not exist in the provided context. The entire baseline response (e.g., "there is no information...") is absent from the GT, resulting in zero precision.
>         * Example: The baseline claims, "there is no data shown for the Zigbee packet error rate." The GT image provides the correct trend for the Zigbee rate, indicating the baseline failed to find it.
>     * **Over-reporting**: This pattern results in low precision scores even when the baseline's core answer is correct. The baseline adds extra, unrequested details, interpretations, or introductory text that is not in the ground-truth contexts.
>         * Example: The baseline correctly identifies most state transitions but misidentifies one and completely misses the "RESET" event, making it an incomplete and thus partially unfaithful description of the diagram.
> * For Completeness, we check all the question-answer pairs which **Claude** fails to answer incompletely **with ground-truth text and image chunks**, we find the following patterns.
>     * **Factual Contradictions**: Same with “faithfulness” analysis, the baseline answer is incomplete because it omits 100% of the correct information from the GT and replaces it with a factually incorrect, "hallucinated" answer.
>     * **Unfaithful Refusals**: Same with “faithfulness” analysis, the baseline answer omits the entire GT answer by incorrectly claiming the information does not exist in the provided context.
>     * **Partial Omissions**: The baseline successfully provides some of the correct information from the GT but omits other key components, details, or steps.
>         * Example:
>             * Question: Which logistics providers are utilized by the company for managing shipment tasks according to the goal framework?
>             * Analysis: The baseline correctly identifies "DHL and UPS" but omits "sonic air," which was also present in the GT, leading to a recall score of 0.66.
>
> In detail, these errors are due to:
> * Image Detail Capture:
>     * Chart/Table Misreading: The model fails to correctly extract a specific number, word, or symbol from a dense chart or table.
>     * Spatial Ambiguity: The model struggles with overlapping, incomplete, or geometrically similar shapes, leading it to misidentify an object.
>     * Noise/Resolution: Low-quality, blurry, or "noisy" images make accurate detail capture difficult, leading to incorrect extractions.
> * Contextual Reasoning:
>     * Factual Inaccuracies: The model "sees" all the data points in a chart but fails to synthesize them to understand the trend. It might see "Sales" and "Professionals" but incorrectly conclude which had the most "unsatisfactory" ratings because it failed to correctly link the labels to the bar heights.
> * Failure in Cross-Modal Synthesis: The GT answer often requires the model to find two or more separate pieces of information and then synthesize them into a single, new conclusion.
>     * Factual Contradictions: The model "hallucinates" a connection that isn't there. It correctly sees Fact A in the text and Fact B in the chart and incorrectly reasons that A causes B, leading to a factually inaccurate statement.
>
>
> Although SoTA models are still rapidly evolving, we are planning and continue to conduct in-depth analyses of different models and aim to provide unified guidance to the community.

---

> ### Author Response · Authors · 2025-11-26
> **Friendly Reminder**
>
> We thank you for your constructive feedback. In response, we also have added **Long-Context VQA experiments**, as some reviewers requested clarification on the broader utility of our dataset. The new results show clear gains across domains: **GPT-5 achieves the highest Precision (0.658) and Claude Sonnet 4.5 achieves the highest Recall (0.708)**, outperforming Gemini 2.5 Pro, and Qwen2.5-VL-32B. These improvements demonstrate that our dataset is not only effective for standard RAG but also highly beneficial for long-context VQA scenarios. We kindly invite you to check the full VQA results in the comment in the main thread.
>
> We encourage you to take a look at the updated results and revisions. If these additions address your concerns, we would greatly appreciate a reconsideration of the scores, and we are happy to clarify any remaining questions. Thank you again for your time and helpful feedback.

---

### Official Review · Reviewer_jCxW · 2025-11-01

**Soundness:** 2
**Presentation:** 2
**Contribution:** 2
**Rating:** 4
**Confidence:** 2

**Summary:**

This paper presents UniDoc, a unified benchmark designed to evaluate multimodal document understanding across diverse input forms, e.g., scanned pages, forms, and charts. UniDoc aggregates and harmonizes multiple existing datasets into a standardized schema, supporting different types of tasks. The benchmark construction is illustrated, and relevant experiments are executed.

**Strengths:**

- The benchmark is inclusive. As shown in Table 1, UniDoc contains the tasks from multiple domains, and it has the largest number of queries and pages of documents.
- The dataset curation section is clear to some extent, and the further release and deployment seem promising.

**Weaknesses:**

- The paper mostly reuses and reprocesses existing datasets, offering engineering unification rather than new data collection or annotation.
- 20% generated QA pairs are validated by annotators or experts. More can increase the credibility of the benchmark.
- A clear task definition should be formally expressed along with the proposed benchmark.
- The tested baseline model scope should be enlarged to increase the credibility of the proposed benchmark. So far, only 4 main models are included in the paper. Although tables 4 and 5 involve 6 models, this is still not adequate.

**Questions:**

In addition to the weakness, other questions are listed
- Is it necessary to measure the difficulty of tasks when unifying different datasets?
- Is there any plan for including reasoning chain annotations to assess step-by-step interpretability?

**Details Of Ethics Concerns:**

Different source datasets may have distinct licensing terms. Clarity is needed for redistribution under a unified release.

---

> ### Author Response · Authors · 2025-11-18
> **Beyond Reuse: A Curated and Newly Annotated Benchmark**
>
> Reviewer: _The paper mostly reuses and reprocesses existing datasets, offering engineering unification rather than new data collection or annotation._
>
> **Our Response:**
>
> We agree that UniDoc-Bench builds on an existing PDF corpus (PDFA), but our benchmark is not a trivial reprocessing of that data. PDFA is a large, unlabeled, and noisy collection of heterogeneous PDFs, making it unsuitable for use as a RAG database without substantial processing.  In this work, we (1) develop a classification-based tagging and filtering pipeline that processes **460,645** raw PDFA PDFs to curate a high-quality, domain-balanced multimodal document database of only **12,874 PDFs (2.8%) with 70k pages**, with explicit metadata such as domain, subdomain, modality coverage, image quality, and text proportion; and (2) construct and annotate 1,600 new multimodal QA pairs that did not previously exist, each grounded in text, tables and figures.
>
> These QA pairs are generated via a structured synthesis pipeline and subsequently refined for human intent, self-containment, and evidence grounding, and verified by three independent annotators. Together, this curation and annotation effort yields a new benchmark specifically tailored to document-centric MM-RAG, rather than merely unifying existing labeled datasets.
>
> Furthermore, our **curation pipeline has been verified by human annotators** to create high-quality QA pairs from any document set, so users can easily build their own MM-RAG evaluation data for their specific domains and needs.

---

> ### Author Response · Authors · 2025-11-18
> **Significant Human Effort Behind Our 20% Validation**
>
> Reviewer: _"20% generated QA pairs are validated by annotators or experts. More can increase the credibility of the benchmark."_
>
> **Our Response:**
>
> We appreciate this suggestion and agree that increasing the proportion of human-validated QA pairs can further strengthen any benchmark. In our case, validating 20% of UniDoc-Bench (320 QA pairs) was already a **highly resource-intensive effort**. This corresponds to an estimated 640 annotator-hours: 320 questions (20% of 1,600) × 4 annotators (3 independent annotators + 1 expert reviewer) × 30 minutes per question ≈ **38,400 minutes in total**.
>
> This time reflects the complexity of the task. For each sampled QA pair, annotators had to carefully read the **entire document** (not just the retrieved chunks) to verify multiple criteria: (1) factuality, i.e., the question is fully supported by the source document; (2) answer completeness, i.e., no necessary information is omitted anywhere in the document (so the annotators have to read the entire doc); and (3) grounding, i.e., every cited evidence segment is genuinely required to answer the question, which also meant carefully inspecting all text, tables, and images.
>
> This rigorous, multi-annotator validation serves two purposes: (1) it confirms the high quality of UniDoc-Bench itself and, importantly, (2) it proves that our data-synthesis pipeline can reliably create accurate, well-grounded multimodal QA pairs. Since the pipeline is modular and fully described, users can easily validate more items or use the same process to build new MM-RAG datasets from their own documents.

---

> ### Author Response · Authors · 2025-11-18
> **Additional Advanced Models Evaluated for UniDoc-Bench (pt.1)**
>
> Reviewer: _"The tested baseline model scope should be enlarged to increase the credibility of the proposed benchmark. So far, only 4 main models are included in the paper. Although tables 4 and 5 involve 6 models, this is still not adequate."_
>
> **Our Response:**
>
> Thank you for the suggestion. We agree that adding results from more state-of-the-art models will strengthen our paper. Using (1) the best top-k values (10 and 20) from Table 3 and (2) the ground-truth images, tables, and text chunks (GT), we evaluated three additional advanced models: **GPT-5**, **Gemini-2.5-Pro**, and **Claude-4.5-Sonnet**.
> Below are the **Faithfulness** results across different domains.
>
>
> | Domain | Gemini-2.5-Pro (topk=10) | Gemini-2.5-Pro (topk=20) | Gemini-2.5-Pro (GT) | Claude-4.5-Sonnet (topk=10) | Claude-4.5-Sonnet (topk=20) | Claude-4.5-Sonnet (GT) | GPT-5 (topk=10) | GPT-5 (topk=20) | GPT-5 (GT) |
> |--------|--------------------------|---------------------------|---------------------|----------------------------|-----------------------------|------------------------|-----------------|-----------------|------------|
> | Commerce Manufacturing | 0.7327 | 0.7218 | 0.7793 | 0.6844 | 0.6853 | 0.7845 | **0.7624** | **0.7689** | **0.8046** |
> | Construction | 0.6218 | 0.6289 | 0.7146 | 0.6259 | 0.5987 | 0.7154 | **0.6738** | **0.6954** | **0.7251** |
> | Crm | 0.6366 | 0.6644 | 0.7654 | 0.6299 | 0.6506 | 0.7575 | **0.6893** | **0.6968** | **0.8093** |
> | Education | 0.6798 | 0.6796 | 0.7698 | 0.5947 | 0.6004 | 0.7577 | **0.6863** | **0.7192** | **0.7964** |
> | Energy | 0.6973 | 0.7034 | **0.7819** | 0.6873 | 0.6772 | 0.7495 | **0.7353** | **0.7038** | 0.7515 |
> | Finance | 0.6385 | 0.6255 | 0.7161 | 0.6053 | 0.6034 | 0.6990 | **0.6826** | **0.6778** | **0.7334** |
> | Healthcare | 0.6531 | 0.6857 | 0.7594 | 0.5760 | 0.5955 | 0.7272 | **0.6898** | **0.7035** | **0.7694** |
> | Legal | 0.7342 | 0.7289 | 0.7757 | 0.6849 | 0.6698 | 0.7573 | **0.7553** | **0.7369** | **0.8174** |
> | **Average** | 0.6742 | 0.6798 | 0.7578 | 0.6361 | 0.6351 | 0.7435 | **0.7094** | **0.7128** | **0.7759** |
>
> Below are the **Completeness** results across different domains.
>
> | Domain | Gemini-2.5-Pro (topk=10) | Gemini-2.5-Pro (topk=20) | Gemini-2.5-Pro (GT) | Claude-4.5-Sonnet (topk=10) | Claude-4.5-Sonnet (topk=20) | Claude-4.5-Sonnet (GT) | GPT-5 (topk=10) | GPT-5 (topk=20) | GPT-5 (GT) |
> |--------|--------------------------|---------------------------|---------------------|----------------------------|-----------------------------|------------------------|-----------------|-----------------|------------|
> | Commerce Manufacturing | 0.6871 | 0.7130 | 0.7005 | **0.8040** | **0.7977** | **0.8246** | 0.7229 | 0.7369 | 0.7692 |
> | Construction | 0.6481 | 0.6380 | 0.6455 | **0.6977** | **0.7016** | **0.7284** | 0.6407 | 0.6367 | 0.6946 |
> | Crm | 0.6586 | 0.7012 | 0.7511 | **0.7570** | **0.7603** | **0.8157** | 0.6603 | 0.6671 | 0.7707 |
> | Education | 0.6703 | 0.7131 | 0.7135 | **0.7366** | **0.7817** | **0.8196** | 0.6369 | 0.6947 | 0.7651 |
> | Energy | 0.6573 | 0.6801 | 0.6332 | **0.8108** | **0.7777** | **0.7978** | 0.7017 | 0.7048 | 0.7098 |
> | Finance | 0.6738 | 0.6659 | 0.6722 | **0.7753** | **0.7550** | **0.7721** | 0.7079 | 0.7021 | 0.7212 |
> | Healthcare | 0.6884 | 0.6754 | 0.7494 | **0.7762** | **0.7525** | **0.8459** | 0.6699 | 0.6851 | 0.7582 |
> | Legal | 0.7299 | 0.7327 | 0.6949 | **0.7673** | **0.7489** | **0.7902** | 0.7049 | 0.7297 | 0.7554 |
> | **Average** | 0.6767 | 0.6899 | 0.6950 | **0.7656** | **0.7594** | **0.7993** | 0.6806 | 0.6946 | 0.7430 |

---

> > ### Author Response · Authors · 2025-11-18
> > **Additional Advanced Models Evaluated for UniDoc-Bench (pt.2)**
> >
> > Below are the **Faithfulness** results across different question and answer types.
> >
> > | Category | Gemini-2.5-Pro (topk=10) | Gemini-2.5-Pro (topk=20) | Gemini-2.5-Pro (GT) | Claude-4.5-Sonnet (topk=10) | Claude-4.5-Sonnet (topk=20) | Claude-4.5-Sonnet (GT) | GPT-5 (topk=10) | GPT-5 (topk=20) | GPT-5 (GT) |
> > |----------|--------------------------|---------------------------|---------------------|----------------------------|-----------------------------|------------------------|-----------------|-----------------|------------|
> > | **Image Only** | 0.4642 | 0.4724 | 0.5494 | 0.4344 | 0.4361 | 0.5155 | **0.5344** | **0.5480** | **0.6036** |
> > | **Image Plus Text As Answer** | 0.6192 | 0.6152 | 0.6542 | 0.5689 | 0.5734 | 0.6714 | **0.6501** | **0.6419** | **0.7126** |
> > | **Table Required** | 0.7871 | 0.8038 | **0.8689** | 0.7732 | 0.7524 | 0.8577 | **0.8037** | **0.8210** | 0.8390 |
> > | **Text Only** | 0.8050 | 0.8103 | 0.9452 | 0.7489 | 0.7589 | 0.9212 | **0.8315** | **0.8339** | **0.9469** |
> > | **Causal Reasoning** | 0.6847 | 0.7011 | **0.8542** | 0.6232 | 0.6098 | 0.8227 | **0.7488** | **0.7236** | 0.8496 |
> > | **Comparison** | 0.6632 | 0.6609 | 0.6752 | 0.6130 | 0.6286 | 0.6622 | **0.7091** | **0.6968** | **0.7126** |
> > | **Factual Retrieval** | 0.6615 | 0.6700 | 0.7470 | 0.6378 | 0.6382 | 0.7309 | **0.6915** | **0.7120** | **0.7652** |
> > | **Summarization** | 0.6847 | 0.6903 | 0.8048 | 0.6503 | 0.6306 | 0.8207 | **0.6936** | **0.7205** | **0.8339** |
> > | **Average** | 0.6742 | 0.6798 | 0.7578 | 0.6361 | 0.6351 | 0.7435 | **0.7094** | **0.7128** | **0.7759** |
> >
> >
> >
> > Below are the **Completeness** results across different question and answer types.
> >
> > | Category | Gemini-2.5-Pro (topk=10) | Gemini-2.5-Pro (topk=20) | Gemini-2.5-Pro (GT) | Claude-4.5-Sonnet (topk=10) | Claude-4.5-Sonnet (topk=20) | Claude-4.5-Sonnet (GT) | GPT-5 (topk=10) | GPT-5 (topk=20) | GPT-5 (GT) |
> > |----------|--------------------------|---------------------------|---------------------|----------------------------|-----------------------------|------------------------|-----------------|-----------------|------------|
> > | **Image Only** | 0.5688 | 0.5815 | 0.6332 | **0.6252** | 0.6019 | 0.6769 | 0.5909 | **0.6020** | **0.6815** |
> > | **Image Plus Text As Answer** | 0.6075 | 0.6362 | 0.6857 | **0.7081** | **0.7170** | **0.8096** | 0.6013 | 0.5999 | 0.7002 |
> > | **Table Required** | 0.7673 | 0.7730 | 0.7123 | **0.8502** | **0.8542** | **0.8057** | 0.7465 | 0.7741 | 0.7530 |
> > | **Text Only** | 0.7612 | 0.7619 | 0.7605 | **0.8526** | **0.8469** | **0.8940** | 0.7657 | 0.7843 | 0.8380 |
> > | **Causal Reasoning** | 0.6650 | 0.6863 | 0.6832 | **0.7815** | **0.8043** | **0.8601** | 0.6925 | 0.6869 | 0.7562 |
> > | **Comparison** | 0.7058 | 0.7131 | 0.7195 | **0.7951** | **0.7836** | **0.8043** | 0.6906 | 0.6845 | 0.7396 |
> > | **Factual Retrieval** | 0.6693 | 0.6828 | 0.6999 | **0.7266** | **0.7190** | **0.7536** | 0.6703 | 0.6957 | 0.7288 |
> > | **Summarization** | 0.6603 | 0.6658 | 0.6722 | **0.7757** | **0.7673** | **0.8496** | 0.6529 | 0.6856 | 0.7795 |
> > | **Average** | 0.6767 | 0.6899 | 0.6950 | **0.7656** | **0.7594** | **0.7993** | 0.6806 | 0.6946 | 0.7430 |

---

> > > ### Author Response · Authors · 2025-11-18
> > > **Additional Advanced Models Evaluated for UniDoc-Bench (pt.3)**
> > >
> > > * Faithfulness
> > >     - **GPT-5**: **Faithfulness Champion**
> > >       - Highest average Faithfulness (**0.7759 GT**).
> > >       - Most resilient to imperfect retrieval, dropping only **0.0631** from GT to topk=20.
> > >       - Excels on **Text Only**, **Table Required**, and **Causal Reasoning** questions, demonstrating robust reasoning.
> > >
> > >     - **Claude-4.5-Sonnet**: **Faithfulness weaker and retrieval-sensitive**
> > >       - Drops **0.1084** from GT (0.7435) to topk=20 (0.6351).
> > >       - Highly faithful when given perfect data but struggles with retrieved context.
> > >
> > >     - **Gemini-2.5-Pro**: **Solid Faithfulness, but below GPT-5**
> > >       - Average Faithfulness of **0.7578 GT**.
> > >       - Minor improvements from increasing k; generally robust but not the best.
> > >
> > >     - **All models**: Increasing top-k from 10 to 20 shows negligible or slightly negative impact on Faithfulness.
> > >
> > > * Completeness
> > >     - **Claude-4.5-Sonnet**: **Completeness Champion**
> > >       - Highest Completeness score (**0.7993 GT**).
> > >       - Strong across most answer types.
> > >
> > >     - **GPT-5**: **Good Completeness, slightly behind Claude**
> > >       - Average Completeness **0.7430 GT**.
> > >       - Competitive performance, especially for text-heavy questions.
> > >
> > >     - **Gemini-2.5-Pro**: **Limited by model capability**
> > >       - GT score (0.6950) almost identical to topk=20 (0.6899).
> > >       - Providing perfect information does not improve completeness; bottleneck is the model itself.
> > >
> > >     - **All models**: Struggle with **image-based questions**. Example: GPT-5 only 0.6036 (GT) on "Image Only" vs >0.92 on "Text Only". Faithful, multimodal reasoning remains challenging.
> > >
> > > * Model Trade-offs
> > >     - **GPT-5**: **Best for Faithfulness**, strong second for Completeness.
> > >     - **Claude-4.5-Sonnet**: **Best for Completeness**, weaker in Faithfulness.
> > >     - **Gemini-2.5-Pro**: Strong Faithfulness runner-up, weakest Completeness.
> > >     - **Recommendation**: Choose a model based on task priority—**Faithfulness** vs **Completeness**.

---

> ### Author Response · Authors · 2025-11-18
> **Task Difficulty Categorization in UniDoc-Bench**
>
> Reviewer: _"Is it necessary to measure the difficulty of tasks when unifying different datasets?"_
>
> **Our Response:**
>
> We agree that measuring and analyzing task difficulty is important when unifying heterogeneous datasets. In UniDoc-Bench, we measure difficulty across multiple dimensions instead of reducing it to a single number. Concretely, we categorize **question types** into _Fact Retrieval_, _Comparison_, _Summary_, and _Logical reasoning_, and **answer types** into _Text-only_, _Image-only_, _Text+Image_, and _Table-required_. We then report performance across these categories and across domains (see Tab. 5, Sec. 4.3, and App. F), which reveals that different RAG solutions have distinct strengths and weaknesses under different difficulty profiles.
> Moreover, different domains introduce their own challenges (e.g., visually dense scientific figures), which is why we cover 8 common domains and analyze domain-specific behaviors.
>
> In general, it is hard to define a single “global” difficulty score because difficulty is multi-dimensional (question type, answer type, multi-source vs. single-source, domain, etc.). However, our empirical results support a natural ordering, such as **multi-source > single-source and image-only > text+image > text-only / table-required** in terms of difficulty. We see UniDoc-Bench’s fine-grained categorization and analysis as a first step toward systematic difficulty modeling for MM-RAG, and we plan to explore more explicit difficulty metrics as future work.

---

> ### Author Response · Authors · 2025-11-18
> **Future Plans for Reasoning-Chain Annotations**
>
> Reviewer: _"Is there any plan for including reasoning chain annotations to assess step-by-step interpretability?"_
>
> **Our Response:**
>
> We appreciate this suggestion and agree that explicit reasoning-chain annotations are valuable for studying step-by-step interpretability. In our current release, UniDoc-Bench primarily targets a foundational yet challenging capability: requiring a system to locate and aggregate all necessary evidence from multiple modalities (text, tables, figures) to produce a complete answer. This already **implicitly** involves reasoning, especially for our _Logical_ and _Comparison_ question types. But the emphasis is on evidence aggregation rather than multi-step chains.
>
> We aim to enrich UniDoc-Bench with multi-hop questions, where the answer from one step is required to identify the evidence for the next. However, this type of data is very challenging to collect, and we are still in the process of developing it.

---

> ### Author Response · Authors · 2025-11-18
> **Flag For Ethics Review**
>
> **Our Response:**
>
> PDFA is a filtered subset of the SafeDocs corpus **CC-MAIN-2021-31-PDF-UNTRUNCATED**, which is derived from Common Crawl and distributed via Digital Corpora. Its use is therefore governed by **Common Crawl’s Terms of Use** and the **Digital Corpora Terms of Use**, rather than a standalone dataset license.
>
> In UniDoc-Bench, we only add our own metadata and QA annotations, which we release under a **non-commercial research license**, while the underlying PDFs remain subject to the original web-source terms. As our benchmark is intended for research use, we do not anticipate additional licensing concerns beyond complying with those upstream terms.

---

> ### Author Response · Authors · 2025-11-18
> **A Clear Task Definition of Multimodal RAG**
>
> Thanks for this question. Here is the **Task definition** of MMRAG.
>
> Let $\mathcal{D} = \\{d_1, \dots, d_N\\}$ be a corpus of multimodal documents.  Each document $d_i$ is decomposed into a set of segments
>
> $$
> d_i = \\{ s_{i,j}^{(m)} \mid j = 1,\dots, L_i, m \in \mathcal{M} \\},
> $$
> where $\mathcal{M}$ is the set of modalities (e.g., text, table, figure, equation), and $s_{i,j}^{(m)}$ denotes the $j$-th segment of modality $m$ in document $d_i$.
>
> In UniDoc-Bench, each query is **text-only**. We denote a query as
>
> $$
> q \in \mathcal{X}_{\text{text}},
> $$
> and for each query $q$, the benchmark provides a ground-truth answer $a^\star$ and a set of supporting evidence segments
>
> $$
> E^\star \subseteq \bigcup_{i=1}^N d_i,
> $$
> where $E^\star$ may include segments from different modalities (e.g., text paragraphs, tables, and figures).
>
> A MMRAG model consists of:
>
> * a **retriever**
>
>   $$
>   R_\theta : (q, \mathcal{D}) \mapsto E \subseteq \bigcup_{i=1}^N d_i,
>   $$
>   which selects a subset of multimodal segments $E$ as evidence, and
>
> * a **generator**
>
>   $$
>   G_\phi : (q, E) \mapsto \hat{a},
>   $$
>   which produces a free-form answer $\hat{a}$ conditioned on the text query and the retrieved multimodal evidence.
>
> The overall MMRAG system is the composition
>
> $$
> f_{\theta,\phi}(q, \mathcal{D}) = \big(\hat{a}, E\big)
> = \big( G_\phi(q, R_\theta(q, \mathcal{D}))\, R_\theta(q, \mathcal{D}) \big).
> $$
>
>
> Given a benchmark of text queries
>
> $$
> \\{q_k\\}_{k=1}^{K}
> $$
> with ground-truth pairs
>
> $$
> \\{(a_k^\star, E_k^\star)\\}_{k=1}^{K},
> $$
>
> the goal is to learn or evaluate $f_{\theta,\phi}$ such that
>
>
> $$
> \hat{a}_k \approx a_k^\star \quad \text{and} \quad E_k \approx E_k^\star
> $$
>
> according to answer-level metrics (e.g., exact match, F1, LLM-based scoring) and evidence-level metrics (e.g., precision/recall over segments).

---

> ### Author Response · Authors · 2025-11-26
> **Friendly Reminder**
>
> We thank you for your constructive feedback. In response, we also have added **Long-Context VQA experiments**, as some reviewers requested clarification on the broader utility of our dataset. The new results show clear gains across domains: **GPT-5 achieves the highest Precision (0.658) and Claude Sonnet 4.5 achieves the highest Recall (0.708)**, outperforming Gemini 2.5 Pro, and Qwen2.5-VL-32B. These improvements demonstrate that our dataset is not only effective for standard RAG but also highly beneficial for long-context VQA scenarios. We kindly invite you to check the full VQA results in the comment in the main thread.
>
> We encourage reviewers to take a look at the updated results and revisions. If these additions address your concerns, we would greatly appreciate a reconsideration of the scores, and we are happy to clarify any remaining questions. Thank you again for your time and helpful feedback.

---

### Author Response · Authors · 2025-11-26
**Long-Context VQA Experiments**

We conducted additional experiments to evaluate the utility of our dataset for long-context VQA. The results demonstrate consistent gains across domains, highlighting the effectiveness of our dataset beyond standard Multi-modal RAG tasks.

* Faithfulness

| Domain | Qwen2.5-VL-32B | Claude Sonnet 4.5 | Gemini 2.5 Pro | GPT-5 |
|--------|----------|----------|----------|----------|
| Commerce | 0.376 | 0.551 | 0.595 | 0.678 |
| Construction | 0.268 | 0.468 | 0.526 | 0.593 |
| CRM | 0.271 | 0.555 | 0.578 | 0.646 |
| Education | 0.369 | 0.537 | 0.605 | 0.640 |
| Energy | 0.316 | 0.579 | 0.577 | 0.695 |
| Finance | 0.238 | 0.493 | 0.545 | 0.627 |
| Healthcare | 0.370 | 0.464 | 0.549 | 0.654 |
| Legal | 0.243 | 0.493 | 0.614 | 0.734 |
| **Average** | **0.306** | **0.518** | **0.573** | **0.658** |

* Completeness

| Domain | Qwen2.5-VL-32B | Claude Sonnet 4.5 | Gemini 2.5 Pro | GPT-5 |
|--------|----------|----------|----------|----------|
| Commerce | 0.194 | 0.694 | 0.616 | 0.625 |
| Construction | 0.169 | 0.665 | 0.545 | 0.589 |
| CRM | 0.183 | 0.751 | 0.614 | 0.613 |
| Education | 0.296 | 0.725 | 0.604 | 0.604 |
| Energy | 0.189 | 0.718 | 0.589 | 0.651 |
| Finance | 0.182 | 0.733 | 0.577 | 0.610 |
| Healthcare | 0.235 | 0.719 | 0.557 | 0.591 |
| Legal | 0.166 | 0.655 | 0.618 | 0.682 |
| **Average** | **0.202** | **0.708** | **0.590** | **0.621** |

**Key Observations:**

* GPT-5 achieves the highest Precision (0.658) across domains.
* Claude Sonnet 4.5 achieves the highest Recall (0.708), demonstrating strong coverage.
* Both models outperform Gemini 2.5 Pro and Qwen2.5-VL-32B, highlighting the value of our dataset for long-context reasoning tasks.

These results demonstrate that our dataset is not only effective for standard RAG but also highly beneficial for long-context VQA scenarios.

* Faithfulness by Answer Type

| Answer Type | Qwen2.5-VL-32B | Claude Sonnet 4.5 | Gemini 2.5 Pro | GPT-5 |
|--------|----------|----------|----------|----------|
| Image Only | 0.208 | 0.412 | 0.419 | 0.495 |
| Table Required | 0.230 | 0.648 | 0.614 | 0.750 |
| Image + Text | 0.332 | 0.423 | 0.462 | 0.602 |
| Text Only | 0.408 | 0.564 | 0.630 | 0.765 |

* Completeness by Answer Type

| Answer Type | Qwen2.5-VL-32B | Claude Sonnet 4.5 | Gemini 2.5 Pro | GPT-5 |
|--------|----------|----------|----------|----------|
| Image Only | 0.176 | 0.566 | 0.485 | 0.527 |
| Table Required | 0.156 | 0.809 | 0.586 | 0.696 |
| Image + Text | 0.236 | 0.678 | 0.440 | 0.562 |
| Text Only | 0.241 | 0.771 | 0.574 | 0.685 |

Summarization:
* GPT-5 consistently achieves the highest faithfulness across all answer types, especially Text Only (0.765) and Table Required (0.750).
* Claude Sonnet 4.5 leads in completeness for Table Required (0.809) and Text Only (0.771).
* Image Only and Image + Text answer types remain the most challenging for both RAG and VQA tasks, highlighting key areas for future improvement.

---

### Author Response · Authors · 2025-12-04
**Authors' General Response and Summary of Revisions**

We sincerely appreciate the reviewers' time and effort in evaluating our manuscript and providing constructive feedback. The reviewers collectively recognize UniDoc-Bench as a valuable and large-scale benchmark for Multimodal Retrieval-Augmented Generation (MM-RAG), which offers a unified evaluation framework and provides granular insights into multimodal document understanding (Reviewers 7MPR, wtt8, jCxW, 7bma).

In response to the comments, we have substantially strengthened our manuscript and provided extensive new results. We believe these revisions fully address the raised concerns and significantly reinforce the paper's contributions.

### **Summary of Revisions**
1. Expanded Model Scope and Detailed Analysis (Reviewers wtt8, jCxW)
Reviewers' Concern: The paper should include results from more State-of-the-Art (SoTA) models (e.g., GPT, Gemini, Claude) and provide a more detailed comparison of their performance and error types.

* Our Revision:

    * Added New SoTA Models: We evaluated and added comprehensive results for three advanced commercial models: GPT-5, Gemini-2.5-Pro, and Claude-4.5-Sonnet (in addition to the original 4 models). These results cover both Faithfulness and Completeness metrics across 8 domains and 8 question/answer types, under different retrieval settings (top-k=10, top-k=20, and Ground Truth/GT).

    * Detailed Model Trade-offs: Our analysis provides clear guidance:

        * GPT-5 is the Faithfulness Champion (highest average Faithfulness at 0.7759 GT) and is most resilient to imperfect retrieval.

        * Claude-4.5-Sonnet is the Completeness Champion (highest average Completeness at 0.7993 GT).

        * Gemini-2.5-Pro is a strong Faithfulness runner-up but is constrained by its own generation capability for Completeness.

    * Qualitative Error Analysis: We added a detailed qualitative analysis of typical failure patterns for SoTA models (e.g., Factual Inaccuracies, Chart/Table Misreading, False Negatives, and Failure in Cross-Modal Synthesis), providing actionable insights for the community.

2. Dataset Curation and Credibility (Reviewers jCxW, 7bma)
* Our Response:
    * Response: We clarified that the benchmark is highly curated, involving annotating 1,600 new multimodal QA pairs and selecting only 2.8% of the raw PDF corpus.
    * Validation: We detailed the ~640 hours of human effort invested in validating 20% of the QA pairs to ensure high quality and "Human-like Intent," thereby mitigating any potential LLM synthesis bias (by decoupling LLMs and using human-authored templates).

3. Expanded Utility and Task Definition (Reviewers wtt8, jCxW, 7MPR)
* Our Response:
    * Long-Context VQA: New experiments confirmed the dataset's effectiveness for Long-Context VQA scenarios, showing strong performance gains across models.
    * Task Definition: We added a formal, mathematically-expressed definition of the MM-RAG task to provide clarity and precision to the evaluation framework.
    * Unified Evaluation/Open Source: We reiterated that the unified evaluation protocol ensures fair, "apples-to-apples" comparison across RAG systems. We also confirmed our commitment to open-sourcing the evaluation framework and code.

We believe these revisions fully address the reviewers' concerns, and we hope they meet the expectations outlined by reviewers.

Authors

---

### Note · Authors · 2026-01-05

I have read and agree with the venue's withdrawal policy on behalf of myself and my co-authors.